# An LLM chatbot to facilitate primary-to-specialist care transitions: a randomized controlled trial

Patient-facing large language models (LLMs) hold potential to streamline inefficient transitions from primary to specialist care. We developed the preassessment (PreA), an LLM chatbot co-designed with local stakeholders, to perform the general medical consultations for history-taking, preliminary diagnoses, and test ordering that would normally be performed by primary care providers and to generate referral reports for specialists. PreA was tested in a randomized controlled trial involving 111 specialists from 24 medical disciplines across two health centers, where 2,069 patients (1,141 women; 928 men) were randomly assigned to use PreA independently (PreA-only), use it with staff support (PreA-human), or not use it (No-PreA) before specialist consultation. The trial met its primary end points with the PreA-only group showing significantly reduced physician consultation duration (28.7% reduction; $3.14 \pm 2.25$ min) compared to the No-PreA group ($4.41 \pm 2.77$ min; $P < 0.001$), alongside significant improvements in physician-perceived care coordination (mean scores 113.1% increase; $3.69 \pm 0.90$ versus $1.73 \pm 0.95$; $P < 0.001$) and patient-reported communication ease (mean scores 16.0% increase; $3.99 \pm 0.62$ versus $3.44 \pm 0.97$; $P < 0.001$). Equivalent outcomes between the PreA-only and PreA-human groups confirmed the autonomous operation capability. Co-designed PreA outperformed the same model with additional fine-tuning on local dialogues across clinical decision-making domains. Co-design with local stakeholders, compared to passive local data collecting, represents a more effective strategy for deploying LLMs to strengthen health systems and enhance patient-centered care in resource-limited settings. Chinese Clinical Trial Registry identifier: ChiCTR2400094159.

The growing burden of multimorbidity and aging populations has exposed vulnerabilities in healthcare delivery worldwide[1-3]. Health systems face increasing strain from fragmented infrastructure, under-resourced primary care and inefficient triage mechanisms[4,5], challenges that are particularly acute in regions where self-referral practices bypass primary care for direct tertiary hospital access[6-10]. China's health system exemplifies this crisis: according to the 2023 Statistical Bulletin on China's Health Sector Development, hospital visits reached 4.26 billion in 2023 (11.5% annual increase), while only 59.2% of public hospitals offered appointment systems, driving inefficient care-seeking pathways that overwhelm outpatient services. This imposes a dual burden: specialists face patient consultations without referrals[11], leading to prolonged diagnostic timelines[12,13], compromised emotional support[14,15] and elevated professional burnout[16]; concurrently, patients endure protracted waiting times and fragmented care[17,18]. Although interim solutions like nurse-led triage exist, they

✉ e-mail: malibing1984@163.com; hanshasha@pumc.edu.cn

**Fig. 1 | CONSORT flow diagram for the randomized controlled trial.** Flow diagram depicting the participant enrolment, intervention allocation, follow-up and data analysis.

often lack the training for comprehensive patient assessment and chronic disease management[19]. Addressing these systemic inefficiencies in resource-limited settings requires scalable solutions that can transform strained clinical workflows.

Large language models (LLMs) possess transformative potential to re-engineer hospital workflows and address the systemic inefficiencies amplified by escalating demand. However, current applications remain largely confined to support healthcare professionals in controlled settings, for example, responding to patient portal messages[20,21], aiding clinical reasoning in experimental environments[22,23] or improving medical directions in online pharmacies[24], with limited integration into real-time clinical decision-making. Critically, evidence is lacking for LLM chatbots that directly interact with socioeconomically diverse patient populations while supporting both curative and caring aspects of medicine in high-volume clinical environments[25].

Bridging this gap requires overcoming two critical barriers: mitigating the systematic biases that arise when training patient-facing LLMs on local medical dialogues from resource-limited settings[26,27], and establishing real-world evidence of their clinical utility within time-pressured hospital workflows[28–31]. While localized dialogues have enabled specialized applications, from patient-nurse interactions[32] and mental health support[33] to telemedicine service[34], their direct use in resource-limited clinical environments risks replicating existing care deficits. Consequently, a shift toward simulated dialogues curated from standardized medical corpora is underway, moving beyond a reliance on raw local data[23,35]. Yet, the relative utility of co-design versus passive data collecting for meeting clinical needs remains unknown. This omission begs a central question: should LLMs reflect local practices or help reform them? The answer is critical for global health equity, as passively collected local dialogues may codify and even scale systemic inequities, from diagnostic shortcuts to sociocultural biases[26,27].

To bridge the gap between the potential of LLMs and their practical impact in resource-limited settings, we developed PreA (Pre-Assessment), an LLM chatbot (OpenAI; GPT-4.0 mini) for primary-to-specialist care transitions, using a multistakeholder participatory co-design approach[36]. We engaged diverse community and clinical stakeholders, including patients, care partners, community health workers, physicians, nurses and hospital administrators, to shape a tool that addresses real-world clinical and accessibility needs. The final PreA chatbot integrated a patient-facing chatbot with low-literacy accessibility features and a clinical interface that generates specialist referrals and supports evidence-based decision-making under time constraints (Extended Data Fig. 1).

The decision to deploy this co-designed version of PreA was empirically grounded, informed by a previous simulated experiment that directly compared the co-design approach with additionally fine-tuning the same model with local dialogues. We then evaluated the co-designed PreA in a multicenter, pragmatic, randomized controlled trial (RCT) to assess its effectiveness in facilitating primary-to-specialist care transitions.

## Results

### Patient flow and baseline data

The trial was conducted across 24 medical disciplines at two academic tertiary medical centers in western China (The First Affiliated Hospital of Guilin Medical University and the Affiliated Hospital of Gansu Medical University). A total of 2,332 patients and their care partners were evaluated for eligibility, with 194 either opting out or being excluded for various reasons (Fig. 1). This left 2,138 patients who were randomly assigned in 1:1:1 ratio to use PreA independently (PreA-only, $n = 712$), use it with staff support (PreA-human, $n = 713$) or not use it (No-PreA, $n = 713$). Subsequently, 69 patients opted out or were removed for various reasons.

Our final analysis included 2,069 participants (PreA-only, $n = 691$, PreA-human: $n = 689$, No-PreA, $n = 689$). Participants had a mean age of $47.6 \pm 14.6$ years and included 1,141 women (55.1%) and 928 men (44.9%). Most participants (1,620, 78.3%) were patients themselves, with

**Table 1 | Distribution of baseline covariates across three trial groups**

| Characteristic | Total | PreA-only | Pre-human | No-PreA | P value[a] |
|---|---|---|---|---|---|
| **No. of participants** | 2,069 | 691 | 689 | 689 | |
| **Age**, years, mean±s.d. | 47.6±14.6 | 47.2±14.4 | 47.7±14.9 | 47.8±14.6 | 0.717 |
| **Sex,** n (%) | | | | | 0.134 |
| **Female** | 1,141 (55.1) | 382 (55.3) | 361 (52.4) | 398 (57.8) | |
| **Male** | 928 (44.9) | 309 (44.7) | 328 (47.6) | 291 (42.2) | |
| **Ethnicity**, n (%) | | | | | 0.857 |
| **Han** | 1,929 (93.2) | 647 (93.6) | 640 (92.9) | 642 (93.2) | |
| **Other races**[b] | 140 (6.8) | 44 (6.4) | 49 (7.1) | 47 (6.8) | |
| **Participant type**, n (%) | | | | | 0.375 |
| **Patients** | 1,620 (78.3) | 529 (76.6) | 543 (78.8) | 548 (79.5) | |
| **Care partners** | 449 (21.7) | 162 (23.4) | 146 (21.2) | 141 (20.5) | |
| **Education**, n (%) | | | | | 0.956 |
| **Primary school or below** | 313 (15.1) | 100 (14.5) | 108 (15.7) | 105 (15.2) | |
| **High school** | 1,073 (51.9) | 363 (52.5) | 358 (52.0) | 352 (51.1) | |
| **College or above** | 683 (33.0) | 228 (33.0) | 223 (32.4) | 232 (33.7) | |
| **Work status**, n (%) | | | | | 0.526 |
| **Employed** | 1,188 (57.4) | 400 (57.9) | 390 (56.6) | 398 (57.8) | |
| **Retired** | 346 (16.7) | 115 (16.6) | 107 (15.5) | 124 (18.0) | |
| **Unemployed** | 535 (25.9) | 176 (25.5) | 192 (27.9) | 167 (24.2) | |
| **Income**[c], RMB/month, n (%) | | | | | 0.148 |
| **<2,000** | 770 (37.2) | 250 (36.2) | 257 (37.3) | 263 (38.2) | |
| **2,000–5,000** | 845 (40.8) | 296 (42.8) | 292 (42.4) | 257 (37.3) | |
| **>5,000** | 454 (21.9) | 145 (21.0) | 140 (20.3) | 169 (24.5) | |
| **Medical discipline**, n (%) | | | | | 0.951 |
| **Medical** | 1,186 (57.3) | 401 (58.0) | 394 (57.2) | 391 (56.7) | |
| **Surgical** | 558 (27.0) | 178 (25.8) | 186 (27.0) | 194 (28.2) | |
| **Med-surg**[d] | 194 (9.4) | 65 (9.4) | 64 (9.3) | 65 (9.4) | |
| **Pediatric** | 131 (6.3) | 47 (6.8) | 45 (6.5) | 39 (5.7) | |
| **Study setting**, n (%)[e] | | | | | 0.737 |
| **Guilin** | 1,457 (70.4) | 488 (70.6) | 478 (69.4) | 491 (71.3) | |
| **Gansu** | 612 (29.6) | 203 (29.4) | 211 (30.6) | 198 (28.7) | |

[a]P value for statistical significance was calculated among the three interventional groups, using a two-tailed one-way analysis of variance (ANOVA) for continuous variables and a chi-squared test for categorical variables. [b]Other races include Zhuang, Yao and Hui. [c]included income from an office, employment on a full-time, part-time or casual basis, or a pension from former employment. [d]Med-Surg represents the department that provides both medical and surgical interventions for patients. [e]The First Affiliated Hospital of Guilin Medical University, Affiliated Hospital of Gansu Medical College.

the remainder being care partners. Demographically, fewer than half (881, 42.6%) were unemployed or retired, and 770 (37.2%) reported a monthly income below 2,000 RMB. Educational attainment among them was distributed as follows: below primary school (313, 15.1%), high school (1,073, 51.9%) and college or higher (683, 33.0%). 1,186 (57.3%) participants consulted a medical specialty, 558 (27.0%) a surgical specialty, 194 (9.4%) a combined medical-surgical specialty and the remainder consulted pediatrics (Extended Data Table 1). These baseline covariates were well balanced across the three trial arms, with no significant differences in distribution (Table 1).

Patients and their care partners in the PreA-only group spent approximately 3.51 ±1.50 min interacting with PreA, with no significant differences from those in the PreA-human group (3.48 ± 1.49 min; $P = 0.72$). They conducted, on average, no more than ten conversation turns, again with no significant differences between the two groups (PreA-only, 9.10 ± 1.37 versus PreA-human, 9.05 ± 1.26; $P = 0.51$).

In their live clinical workflows, 111 specialist physicians reviewed PreA-generated and control (with age and sex only) referral reports immediately before patient consultations. These physicians spent an average of 0.25 ± 0.08 min reviewing PreA-generated reports (PreA-only and PreA-human), compared to 0.07 ± 0.06 min on control reports from the No-PreA group.

## Outpatient workflow

We blindly assessed the effectiveness of the PreA consultation on outpatient workflows across three trial groups using data from the PreA platform and electronic hospital records. The primary outcome was the duration of the medical consultation between patients and physicians, defined as the time elapsed from when patients started conversing with physicians to the end of the consultation. The PreA-only group had a significantly shorter consultation duration compared to the No-PreA group (PreA-only 3.14 ± 2.25 versus No-PreA 4.41 ± 2.77 min; $P < 0.001$; Fig. 2a), corresponding to a 28.7% (95% CI 22.7–34.8) relative reduction. No significant difference was observed between PreA-only and PreA-human groups (3.17 ± 2.87 min; $P = 0.17$).

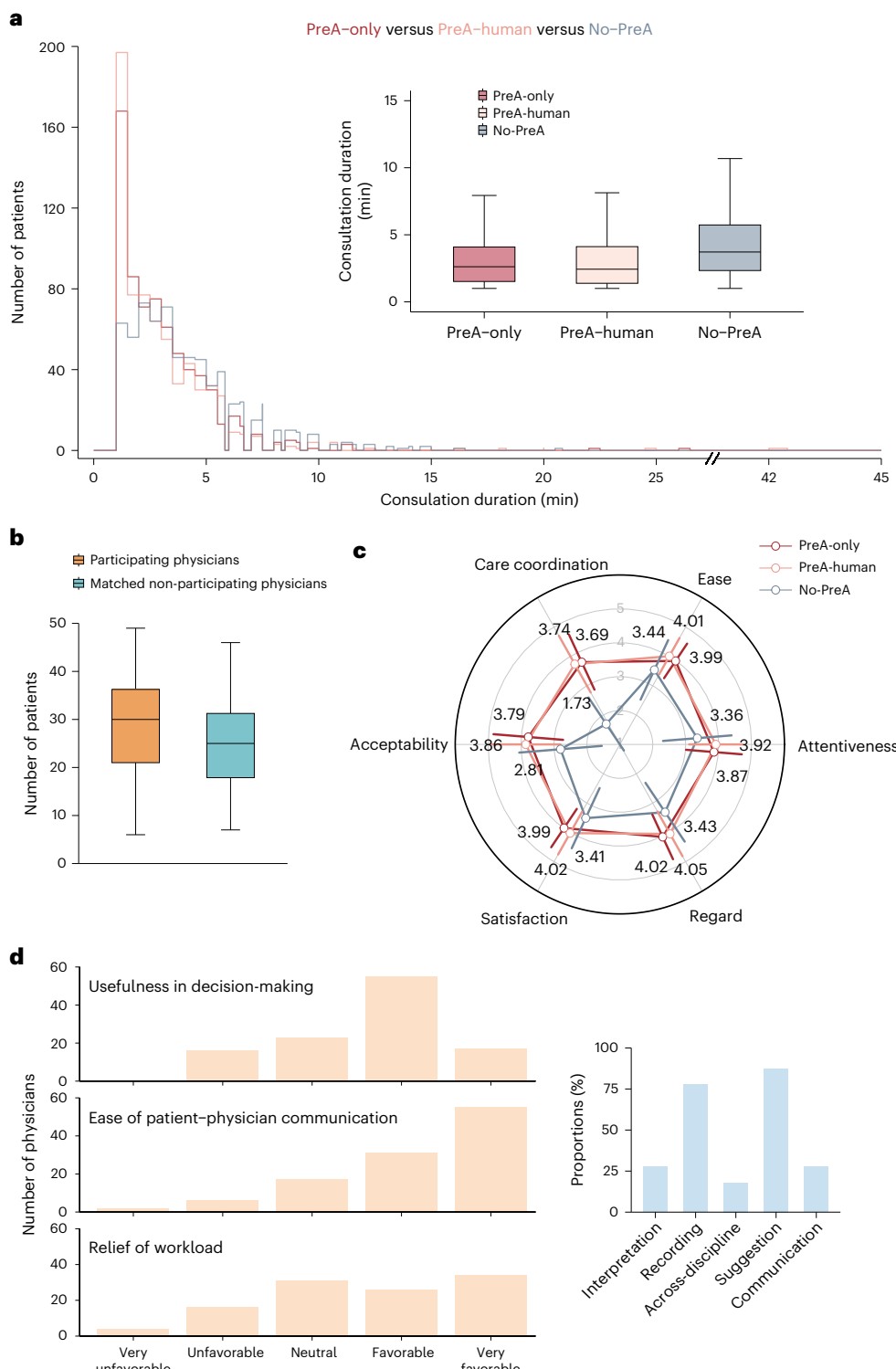

**Fig. 2 | Effects of PreA interventions on outpatient workflow, patient-centeredness and care coordination. a**, Histograms and box plots show the distribution consultation duration across the PreA-only ($n$ = 691 participants), PreA-human ($n$ = 689 participants), and No-PreA ($n$ = 689 participants) groups. The center of the box plot represents the median, with the boundaries representing the first and third quartiles. The whiskers represent the furthest data points from the edge of the box within 1.5 × IQR. **b**, Box plots show patient throughput per shift for participating physicians and nonparticipating physicians based on 80 matched physician pairs. The center of the box plot represents the median, with the boundaries representing the first and third quartiles. The whiskers represent the furthest data points from the edge of the box within 1.5 × IQR. **c**, Radar plots show the patient-centeredness and care coordination metrics across the PreA-only ($n$ = 691 participants), PreA-human

($n$ = 689 participants) and No-PreA ($n$ = 689 participants) groups, with five patient-reported metrics (ease of communication, physician attentiveness, interpersonal regard, patient satisfaction, and future acceptability) and one specialist-rated metric (care coordination). **d**, Bar charts show physician feedback at the end-of-shift questionnaires ($n$ = 111 specialists). The left panel presents ratings of clinical decision support, workload reduction and facilitation of patient–physician communication. The right panel details the most valued features among physicians who rated its usefulness in decision-making as favorable or very favorable. The features include Interpretation (diagnostic report interpretation), Recording (efficient medical history elicitation and documentation), Across-discipline (simultaneous access to multiple specialties), Suggestion (preliminary diagnostic suggestions) and Communication (enhanced patient–physician communication skills).

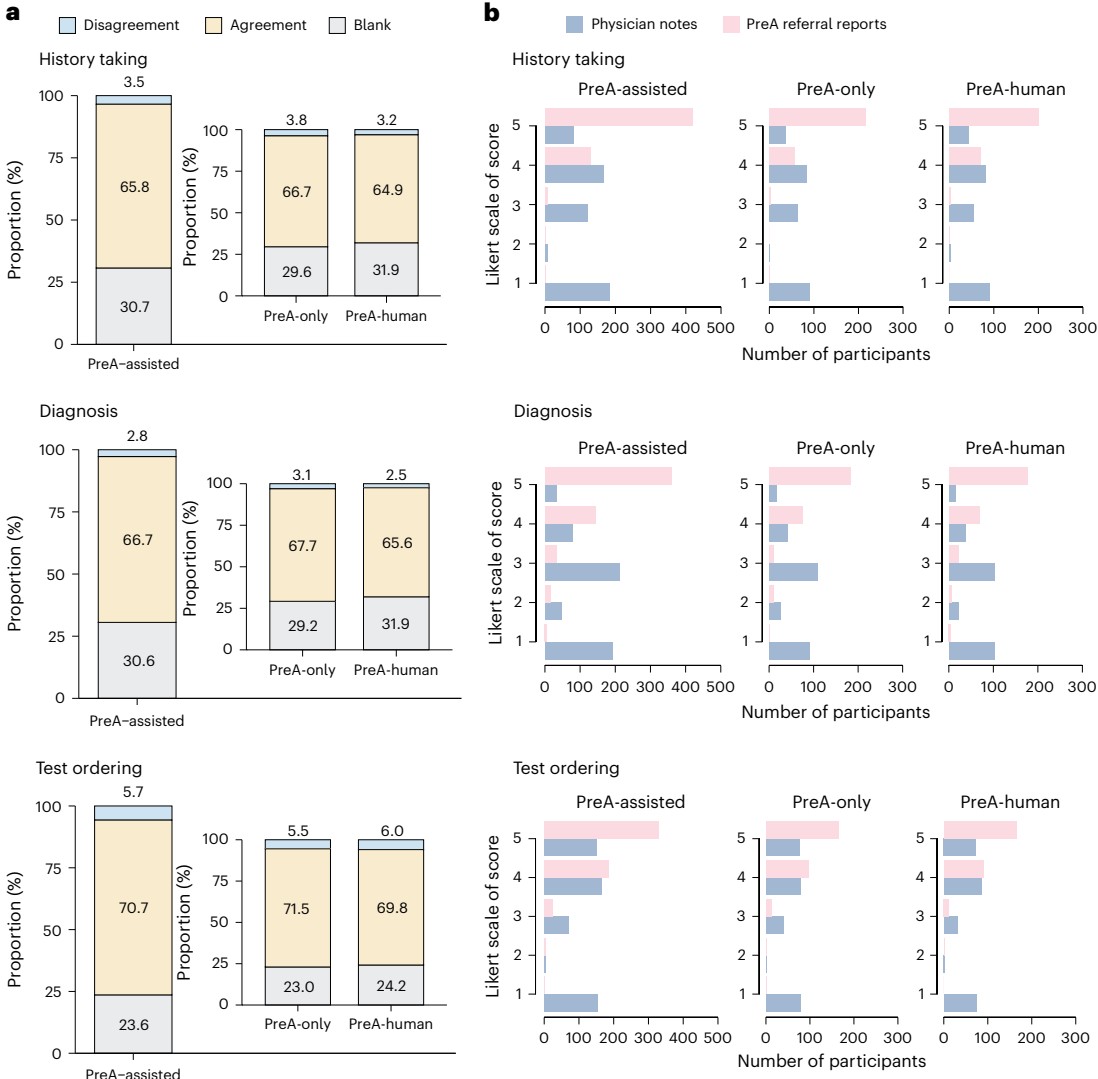

**Fig. 3 | Comparison of PreA-generated referral reports and specialist clinical notes. a**, Agreement analysis between PreA reports and specialist notes. Cases were categorized by level of concordance: agreement (exact match, near-identical content or inclusion of accepted differentials), disagreement or blank (missing physician notes). PreA-assisted group (*n* = 576) combines PreA-only

(*n* = 291) and PreA-human (*n* = 285) cases. **b**, Quality assessment of PreA reports versus specialist notes. The data represent the distribution of expert-evaluated quality across all available cases, including those with blank physician notes. PreA-assisted group (*n* = 1,152 samples) combines PreA-only (*n* = 582) and PreA-human (*n* = 570) samples.

The secondary outcome is the number of patients per clinical shift and the exploratory outcome is patients' waiting time. To evaluate the impact of PreA on the physician workload, we employed a matched-pairs analysis, comparing patients of participating physicians with those of nonparticipating physicians, matched on medical specialty, physician work shift timing and professional title. Patients of participating physicians cared for significantly more patients (28.54 ± 9.58) per shift compared to matched nonparticipating physicians (24.76 ± 9.42, *P* = 0.005; relative increase 15.3% (3.4–27.2); Fig. 2b). Given that participating physicians were exposed to PreA-only or PreA-human patients at a maximum frequency of two-thirds, this increase might represent a conservative estimate; on the other hand, physician could strategically control their workflow, thereby the actual impact of PreA on physician workload could be either more pronounced or less pronounced with universal PreA adoption. Despite caring for more patients, patients of participating physicians experienced similar waiting times compared to those of matched nonparticipating physicians (participating 33.54 ± 38.83 min versus nonparticipating 34.65 ± 36.92 min; *P* = 0.37).

**Patient-centeredness and care coordination**
Outcomes here were self-reported by unmasked participants in the RCT (except for physicians masked between PreA-only and PreA-human arms) and measured using the prespecific survey questionnaire based on five-point Likert scales (Supplementary Tables 1 and 2). Patients and care partners in the PreA-only group reported significantly improved consultation experiences compared to the No-PreA group across the primary outcome, ease of communication: 3.99 ± 0.62 versus 3.44 ± 0.97; *P* < 0.001; relative increase 16.0%, 95% CI 13.5–18.5) and the four secondary outcomes: perceived physician attentiveness: 3.87 ± 0.85 versus 3.36 ± 1.04; *P* < 0.001; relative increase 15.1%, 95% CI 12.1–18.1), interpersonal regard (4.02 ± 0.73 versus 3.43 ± 1.05; *P* < 0.001; relative increase 17.2%, 95% CI 14.4–20.0), patient satisfaction (3.99 ± 0.69 versus 3.41 ± 0.98; *P* < 0.001; relative increase 17.0%, 95% CI 14.3–19.6) and future acceptability (3.79 ± 1.06 versus 2.81 ± 1.26; *P* < 0.001; relative increase 34.7%, 95% CI 30.4–39.1; Fig. 3a). No significant differences were found between the masked PreA-only and PreA-human groups across these dimensions (Extended Data Table 2).

For the primary outcome of referrals in facilitating specialist care, physicians reported a significantly higher value for PreA referral reports compared to the usual one (Care coordination: PreA-only $3.69 \pm 0.90$ versus No-PreA $1.73 \pm 0.95$; $P < 0.001$; Fig. 3b), corresponding to a 113.1% (95% CI 107.4–118.7) relative increase. No significant difference in perceived value was observed between the masked PreA-only and PreA-human groups ($P = 0.45$). At the end of each working shift, physicians provided feedback on the secondary outcomes, including the usefulness of PreA in their clinical decision-making (Fig. 3c). A majority reported PreA to be useful or very useful (64.9%, 72 of 111); among them, preliminary diagnostic suggestions (87.5%, 63 of 72) and efficient medical history acquisition (77.8%, 56 of 72) were identified as the most valuable features. Furthermore, 77.5% (86 of 111) believed it enhanced patient–physician communication (favorable or very favorable), while 54.1% (60 of 111) of physicians perceived PreA as a tool for reducing workload.

### Demographic and socioeconomic differences

Prespecified subgroup analyses, stratified by demographic and clinical characteristics, demonstrated consistent reductions in consultation duration. Notably, these reductions were observed across age groups, sex, educational attainment, work status, income levels, medical disciplines (medical medicine, surgery, mix of medical medicine and surgery, pediatrics), study sites (Guilin/Gansu) and participant type (patients/care partners), with PreA-only showing significant reductions compared to No-PreA, and no significant differences compared to PreA-human (Extended Data Figs. 2–5).

However, patient experience outcomes exhibited some variability across subgroups. While the PreA-only group generally reported superior consultation experiences compared to No-PreA, this effect was not uniformly observed. Specifically, high-income participants and those attending pediatric departments did not report significant differences in perceived physician attentiveness between the PreA-only and No-PreA groups (Extended Data Figs. 3c and 4).

### Physician clinical decision-making

Concerns regarding automation bias and anchoring, as in experimental contexts[21], suggest clinicians may directly adopt LLM-generated assessments, potentially bypassing their clinical reasoning. To investigate this in our real-world trial, we examined whether physicians' clinical notes from the PreA-assisted groups exhibited distinct characteristics from those in the No-PreA group, as per the prespecified analysis.

Classification analysis yielded near-random discriminability (F1 score 0.57; $P = 0.81$; $\Delta F1 < 0.02$). This absence of systematic separability in the feature space of clinical notes provides compelling evidence against the direct adoption of LLM-generated content in this real-world clinical context. We further investigated this finding across five clinical domains of history-taking, physical examination, diagnosis, test ordering and treatment plans. Consistent with the overall findings, no significant difference was observed between the PreA-only and No-PreA groups (or PreA-only and PreA-human groups) in the five domains ($P = 0.10$–$0.90$; Extended Data Table 3). These findings collectively suggest that PreA-assisted medical consultation did not introduce detectable, systematic alterations in physician decision-making, either overall or within specific clinical domains.

### Quality of referral

We performed a blind post hoc analysis comparing PreA referral reports to the subsequent physician clinical notes among the PreA-assisted groups. PreA-generated reports exhibited substantial agreement (exact match, near-identical content or inclusion of accepted differentials) with physician notes in 65.8% (95% CI 61.8–69.6) of history-taking, 66.7% (95% CI 62.7–70.4) of diagnoses, and 70.7% (95% CI 66.8–74.2) of test ordering recommendations (Fig. 3a). PreA reports show disagreement in only 2.8% to 5.7% of cases, while the remaining physician notes were

absent that precluded direct comparison. Among cases exhibiting agreement or where physician notes were blank, PreA reports were rated significantly higher quality than physician notes in terms of completeness, appropriateness and clinical relevance across history-taking (PreA $4.73 \pm 0.50$ versus physician notes $2.93 \pm 1.49$), diagnosis (PreA $4.49 \pm 0.82$ versus physician notes $2.49 \pm 1.25$), and test ordering (PreA $4.55 \pm 0.63$ versus physician notes $3.28 \pm 1.57$; Fig. 3b). Intergroup analysis (PreA-only versus PreA-human) revealed no statistically significant differences in agreement rates and quality scores across all assessed domains.

### Comparative performance of development strategies

The choice of a co-designed chatbot for the RCT was informed by a previous simulated experiment that directly compared this approach against fine-tuning with local dialogues. For this experiment, we compiled a de-identified audio corpus of 515 patient–physician scenarios (199,145 Chinese words) collected across rural clinics and urban community health centers within the same 11 provinces as the co-design process. This dataset comprised general medical consultation interactions in geographically and socioeconomically diverse settings, with 51.7% (266 of 515) from rural areas and 77.9% (401 of 515) from the low-income regions. Mean consultation durations ranged from 1.55 to 3.98 min, and interaction lengths spanned 226.90 to 546.00 Chinese words per scenario (Fig. 4a).

The co-designed model achieved significantly higher-quality rating scores than the data-tuned counterpart (the co-designed model further fine-tuned on the primary care dialogues) group across all domains: history-taking (without data-tuned $4.56 \pm 0.65$ versus data-tuned $3.86 \pm 0.81$; $P < 0.001$; Fig. 4b), diagnosis (without data-tuned $4.67 \pm 0.55$ versus data-tuned $2.47 \pm 1.44$; $P < 0.001$) and testing order (without data-tuned $4.23 \pm 1.09$ versus data-tuned $2.21 \pm 1.12$; $P < 0.001$). Notably, the data-tuned model replicated systemic inefficiencies observed in real-world primary care, including omitting guideline-recommended history elements and demographic elements (for example, patient age and sex), and failing to provide appropriate tests and diagnoses (Supplementary Table 3). Mirroring real-world clinician patterns, the data-tuned model exhibited suboptimal adherence to diagnostic guidelines, failing to provide diagnoses (30.0%, 90 of 300) or suggest testing (39.3%, 118 of 300) when needed. Additionally, the data-tuned model mimicked an unfriendly tone similar to that of human clinicians.

## Discussion

We developed and evaluated PreA, a co-designed LLM-based chatbot that streamlines primary-to-specialist care transitions by preparing patients for consultations and generating preconsultation referrals to specialists. In a pragmatic, multicenter RCT in China, PreA improved both operational efficiency and patient-centered care delivery in high-volume hospital settings compared to usual practice. The findings provide preliminary evidence for the clinical utility of co-designed LLMs within time-constrained clinical workflows, suggesting that co-design with local stakeholders is an effective strategy for deploying LLMs into clinical practice.

The trial demonstrated that PreA enhanced both efficiency and patient-centeredness (a dual benefit rarely achieved in previous LLM deployments)[20–22]. Specialist physicians who received PreA-generated referral reports reduced their average consultation time by 28.7%, indicating that the tool enabled faster synthesis of clinical narratives and supported time-intensive decision-making. Indeed, the majority of specialists endorsed PreA's utility for rapid clinical synthesis, particularly valuing its preliminary diagnostic suggestions and medical history acquisition, which aligns with a recent qualitative investigation on physician views[37]. This efficiency gain, which could expand patient access or improve care quality, is particularly transformative in overloaded health systems where consultation lengths rank among the shortest

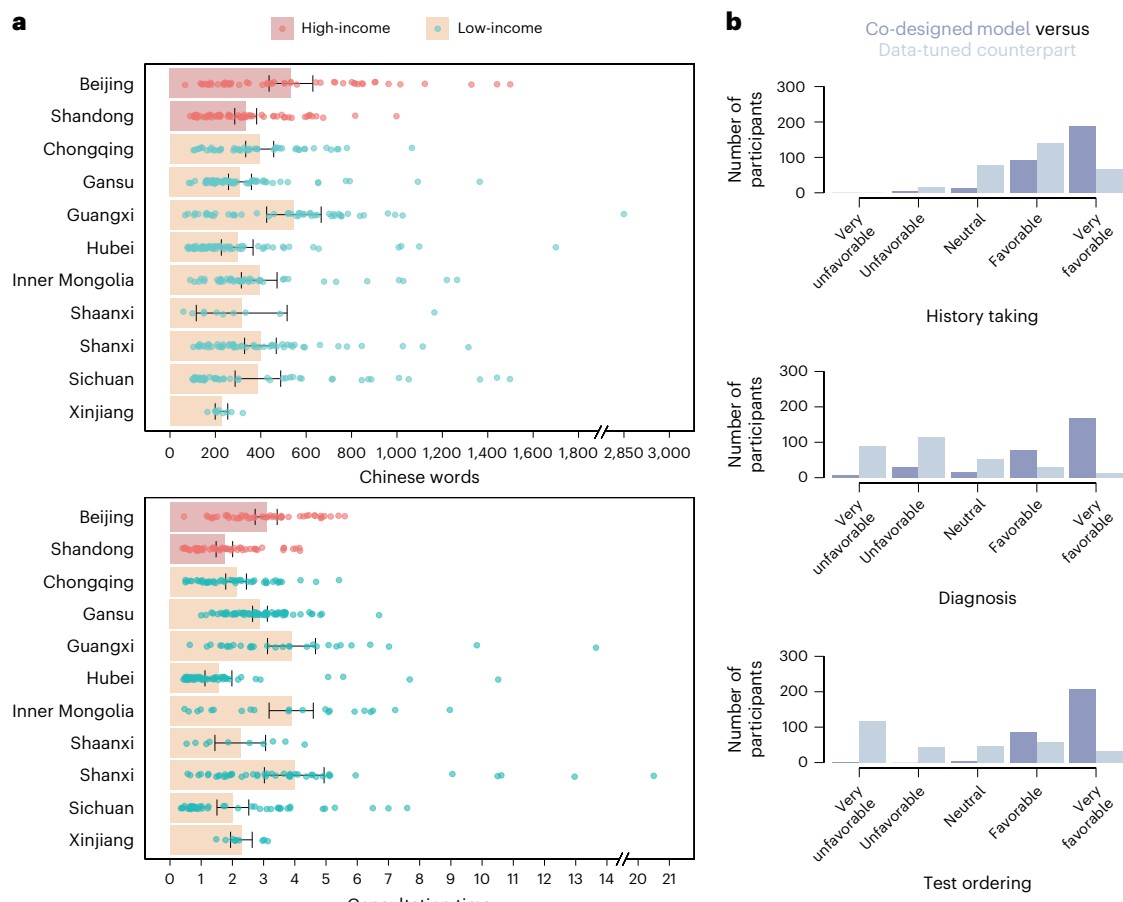

**Fig. 4 | Local dialogue characteristics and model performance comparison.**
**a**, Geographic distribution and characteristics of the de-identified audio corpus comprising 515 patient–physician scenarios collected from the 11 provinces where local stakeholders participated in the co-design process. Provinces are categorized by income levels (high/low). Bar height represents the mean value, dots indicate individual data points, and error bars show 95% CIs. **b**, Quality score distributions for comparing co-designed PreA ($n = 300$ samples) with its local data-tuned counterpart ($n = 300$ samples), the same co-designed base model further fine-tuned on the primary care dialogues, across clinical evaluation domains.

worldwide[38]. Notably, this efficiency gain did not compromise (instead enhanced) both the cure-oriented and care-oriented medicine[39,40], with physicians reporting improved care coordination and patients perceiving a more patient-centered experience. These efficiency cascades help to address core health system constraints identified in our co-design process, suggesting PreA's potential applicability in other health systems facing similar inefficiencies.

The operational autonomy demonstrated by PreA, as evidenced by the equivalent performance of the PreA-only and PreA-human groups, carries important implications for scalability and cost-effectiveness for resource-constrained health systems[41]. Our matched-pair analysis revealed increased patient throughput per clinical shift even under partial PreA adoption, suggesting multiplicative system-level benefits when LLMs streamline preconsultation workflows. In resource-limited settings, such efficiency gains may substantially improve healthcare access, enhancing care equity. Additionally, the higher-quality scores of PreA reports position them as patient-specific templates that could alleviate the burden of clinical documentation. Future large-scale studies are needed to validate these potential benefits across diverse health systems.

Our pre-trial ablation studies highlight an essential pathway toward equitable clinical AI: passively training LLMs on simply curated local dialogues risks perpetuating systemic care deficits, whereas participatory co-design could mitigate these risks and better align models with high-quality care objectives. The data-tuned model replication of suboptimal practices mirrors broader concerns that AI models trained on structurally biased clinical data exacerbate inequities in marginalized populations[42,43]. The co-designed PreA model, refined through input from local stakeholders, including patients, care partners, community health workers, primary care physicians and specialist physicians, outperformed the data-tuned model across all clinical domains. These findings underscore the architectural prioritization of local stakeholder agency through co-design over the passive assimilation of potentially biased natural dialogue data, advancing methodological approaches for equitable AI deployment in healthcare.

Our study, alongside a concurrent trial demonstrating the efficacy of a co-designed chatbot for primary care in low-resource communities[36], establishes participatory co-design as a versatile methodology for developing context-specific healthcare chatbots. While both RCTs employed similar co-design approaches, they target distinct clinical needs: while the primary care chatbot prioritized AI health literacy and accessibility for community home use, PreA was optimized for structured referral generation and time-efficient operation within high-volume specialist workflows. The resulting technical architectures and clinical applications consequently diverged, reflecting their distinct co-design processes and stakeholder priorities. These complementary findings demonstrate how co-design principles can be adapted to develop tailored LLM solutions for diverse healthcare contexts, serving various patient populations and clinical objectives.

In contrast to previous research that has often framed LLMs as physician-interaction diagnostic entities[22,27,44–46], our findings show that

a co-design approach, involving the iterative alignment of LLMs with the prioritized needs of local stakeholders, represents the necessary next step, moving beyond technical promise to clinically integrated, equity-focused AI tools[28]. The streamlined integration of PreA's outputs with specialist cognitive workflows resulted in significant reductions in consultation time and enhanced patient experience across demographic and socioeconomic strata. Critically, these findings challenge the prevailing narrative that medical AI tools inherently depersonalize medicine[47], instead positing that co-designed LLM deployments could empower clinicians to prioritize patient-centered care when freed from cognitive burdens. Furthermore, while previous LLM models may have achieved success within narrow, siloed domains[22], PreA's demonstrated cross-disciplinary effectiveness, spanning both surgical and medical specialties, underscores its potential to unify currently fragmented care pathways across medical disciplines[28].

Several limitations warrant consideration when interpreting our findings. The generalizability of our time-reduction findings may be context-dependent, as our study was conducted in high-volume, resource-limited hospital settings. The effectiveness of PreA is intrinsically tied to this environment of high clinical demand and standardized workflows, and validation in diverse healthcare systems is warranted. Furthermore, the single-blinded, pragmatic trial design, while reflecting real-world conditions where patients would naturally know their preconsultation experience, introduces potential performance bias as patients were aware of their group assignment; however, several factors mitigate this concern: the concordance of findings across objective and subjective outcome assessments, the absence of significant differences in clinical documentation across trial arms and the alignment of control group consultation times with established practice patterns.

Although co-design demonstrated advantages over local data fine-tuning for mitigating biases in LLM development, this approach remains constrained by data quality limitations in health resource-limited settings. Future comparative studies should evaluate co-design versus emerging high-quality primary care dialogue datasets to better understand their relative strengths and applications.

The systemic documentation gaps[48], evidenced by missing physician notes, represent both a limitation and an important finding. While our analytical methods account for this missingness, future implementations could leverage PreA reports as documentation aids to address this widespread challenge in high-volume settings. Moreover, while PreA demonstrated potential as a primary-to-specialist care transition aid, its transition into home-based use would represent an optimal future direction that requires addressing systemic barriers, including AI health literacy, connectivity limitations and cross-institutional data sharing, as indicated by other work on co-designing LLMs for primary care settings[36].

This study provides preliminary evidence for integrating patient-facing LLMs into hospital workflows. While larger multicenter trials with longer follow-up are needed to establish sustained benefits, cost-effectiveness and generalizability, our findings mark a significant step forward. The demonstrated improvements in workflow efficiency and patient–physician experience indicate that co-designed chatbots can reallocate clinician effort from routine data processing toward more nuanced and meaningful patient interactions. This work underscores co-design with local stakeholders as an effective strategy for deploying LLMs to strengthen health systems and enhance patient-centered care in resource-limited settings.

## Online content

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

Xinge Tao ⓓ [1,14], Shuya Zhou[1,14], Kai Ding ⓓ [2,3,4,14], Sairan Li[1], Yanzeng Li ⓓ [5,6], Boyou Wu[1,7], Qirui Huang[1,8], Wangyue Chen[1], Muzi Shen[1], En Meng[1], Xiaowang Chen[9], Hong Hu[10], Jinchao Zhang ⓓ [11], Jie Zhou ⓓ [11], Lei Zou[6], Libing Ma ⓓ [2,3,4] ✉ & Shasha Han ⓓ [1,12,13] ✉

[1]School of Population Medicine and Public Health, Chinese Academy of Medical Sciences & Peking Union Medical College, Beijing, China. [2]Department of Respiratory and Critical Care Medicine, Center of Respiratory Medicine, The First Affiliated Hospital of Guilin Medical University, Guilin, China. [3]Key Laboratory of Basic Research in Respiratory Diseases, Health Commission of Guangxi Zhuang Autonomous Region, Guilin, China. [4]Key Laboratory of Respiratory Diseases, Education Department of Guangxi Zhuang Autonomous Region, Guilin, China. [5]Institute of Artificial Intelligence and Future Networks, Beijing Normal University, Zhuhai, China. [6]Wangxuan Institute of Computer Technology, Peking University, Beijing, China. [7]School of Science, Harbin Institute of Technology, Weihai, China. [8]School of Statistics, East China Normal University, Shanghai, China. [9]Department of Information Technology, The First Affiliated Hospital of Guilin Medical University, Guilin, China. [10]Department of Endocrinology, Affiliated Hospital of Gansu Medical College, Pingliang, China. [11]Pattern Recognition Center, WeChat AI, Tencent Inc, Beijing, China. [12]State Key Laboratory of Respiratory Health and Multimorbidity, Beijing, China. [13]Key Laboratory of Pathogen Infection Prevention and Control (Peking Union Medical College), Ministry of Education, Beijing, China. [14]These authors contributed equally: Xinge Tao, Shuya Zhou, Kai Ding. ✉e-mail: malibing1984@163.com; hanshasha@pumc.edu.cn

## Methods

### Ethics approval

The Chinese Academy of Medical Sciences and Peking Union Medical College and the local medical ethics committee of the First Affiliated Hospital of Guilin Medical University approved the study. The institutional review boards of the Affiliated Hospital of Gansu Medical College approved the study protocol based on their review and the approval from the medical ethics committee of the First Affiliated Hospital of Guilin Medical University. The trial followed the Declaration of Helsinki and the International Conference of Harmonization Guidelines for Good Clinical Practice. We obtained informed consent from all participants in this study. All participants were informed that this was an exploratory experiment, and the results should not be interpreted as direct guidance for clinical interventions at this stage. This study implemented stringent data protection measures, ensuring that all data were anonymized and encrypted to protect privacy. This trial is registered at the Chinese Clinical Trial Registry (identifier ChiCTR2400094159).

### Co-designed architecture and clinical integration

PreA's architecture, derived from the co-design with local stakeholders, integrates a patient-facing chatbot and a clinician interface (Extended Data Fig. 1a). The patient interface collects medical history via voice or text, while the clinical interface generates structured referral reports. Within the consultation workflow, patients or their designated caregivers interacted with PreA first; it then generated a referral (Supplementary Table 4) for specialists to review before their standard consultation.

Co-design workshops revealed that standard clinical documentation for common conditions often lacks the granularity for personalized care and omits critical patient-specific details like pre-existing comorbidities. As such, PreA referral reports were intentionally designed to bridge this gap by synthesizing comprehensive, patient-specific information to facilitate rapid documentation and diagnostic decision-making. The architecture was also engineered to support both multidisciplinary consultation (prioritized by primary care physicians) and evidence-based diagnostic reasoning (emphasized by specialist physicians).

PreA's consultation logic underwent a two-cycle co-refinement process to achieve broader utility across diverse socioeconomic patient populations and adherence to World Health Organization (WHO) guidelines for equitable AI deployment (Extended Data Fig. 1b). The first cycle involved adversarial testing with 120 patients and caregivers, 36 community health workers, 15 physicians and 38 nurses from urban and rural areas across 11 provinces (Beijing, Chongqing, Gansu, Hubei, Shaanxi, Shandong, Shanxi, Sichuan, Guangxi, Inner Mongolia and Xinjiang). This participatory refinement enhanced real-world contextualization and mitigated potential disparities in health literacy and workflow integration[49]. The second cycle employed a virtual patient simulation, specifically modeling low-health-literacy interactions to further optimize the model against co-designed evaluation metrics. Subsequent sections provide further methodological details.

### Model development

**Patient-facing chatbot.** The patient-facing chatbot employs a two-stage clinical reasoning model: inquiry and conclusion. During the inquiry stage, the model was trained to conduct active, multiturn dialogues to gather comprehensive health-related information, adhering to standard guidelines on general medical consultation. In the conclusion stage, the model generated 1–3 differential relevant diagnostic possibilities, each with supporting and refuting evidence to enhance diagnostic transparency and mitigate cognitive anchoring risks[50,51].

**Specialist physician interface.** PreA was configured to generate a referral report for primary-to-care transitions. The report included patient demographics, medical history, chief complaints, symptoms, family history, suggested investigations, preliminary diagnoses, treatment recommendations and a brief summary aligned with clinical reasoning documentation assessment tools (Supplementary Table 4).

**Accessibility and clinical utility.** To ensure accessibility, the platform supports shared access for patients and their caregivers[52]. An LLM-driven agent performs real-time intention analysis to facilitate empathetic communication and simplify language for low-literacy users, with outputs formatted as JSON for streamlined processing.

To improve clinical utility under time constraints, the model was optimized to balance comprehensive data gathering with clinical time constraints, targeting 8–10 conversational turns based on local stakeholder feedback. Primary care physician input drove the incorporation of high-yield inquiry strategies, which in pilot testing reduced consultation times by approximately half (within 4 min).

**Human interaction refinement.** In the adversarial stakeholder testing cycle, we employed prompt augmentation and agent techniques to refine the model, aligning the chatbot with WHO guidelines for ethical AI in primary care while preserving clinical validity[53]. An evaluation panel consisting of community and clinical stakeholders and one AI-ethics-trained graduate student, conducted iterative feedback cycles, focusing on mitigating harmful, biased or noncompliant outputs via adversarial testing.

**Virtual patient interaction refinement.** In the simulation-based refinement cycle, we used bidirectional exchanges between PreA and a synthetic patient agent to enhance consultation quality. We synthesized 600 virtual patient profiles using LLMs grounded in real-world cases; 50% (n = 300) required interdisciplinary consultation to reflect complex care needs. Five board-certified clinicians validated all profiles for medical plausibility and completeness (achieving 5 of 5 consensus).

The patient agent was built on a knowledge graph architecture[54], formalizing patient attributes (demographics, medical history and disease states) as interconnected nodes. The agent was further instructed to emulate common consultation challenges identified by community stakeholders in the first cycle. Interactions concluded automatically upon patient acknowledgment or after ten unresolved inquiry cycles. We randomly chose 300 profiles for refinement and reserved the remainder for comparative simulation studies.

**Evaluation metrics.** The co-design process identified five consultation quality domains for refinement: efficiency (meeting the patient's demanding time lengths), needs identification (accurate recognition of patient concerns), clarity (concise and clear inquiries and responses), comprehensiveness (thoroughness of information) and friendliness (a respectful and empathetic tone).

PreA's performance was rated across these metrics by a panel of five experts (two primary care physicians, two specialists (one in internal medicine and the other in surgical medicine) and one AI-ethics-trained graduate student). Separately, two primary care physicians assessed referral reports for completeness, appropriateness and clinical relevance using a co-designed, five-point Likert scale (Supplementary Table 5). Scores below 3 triggered further iterative refinement.

### Comparative simulation study with virtual patients

We collected audio recordings of primary care consultations from rural clinics and urban community health centers across the 11 Chinese provinces. Provinces were categorized as low-income and high-income based on whether per capita disposable income was below or above the national average (National Bureau of Statistics of China). Local co-design team members who live in these areas manually calibrated the transcripts to ensure validity, as the raw data contained noisy, ambiguous language, interruptions, ungrammatical utterances,

nonclinical discourse and implicit references to physical examinations. All conversational data collected was rigorously de-identified in compliance with relevant regulatory standards (HIPAA) before data analysis.

We conducted a comparative simulation study to evaluate the incremental utility of integrating these localized dialogues. Two model variants were compared: the co-designed PreA model and a local data-tuned counterpart, created by fine-tuning the PreA model (OpenAI; ChatGPT-4.0 mini) on the processed primary care dialogues. Notably, the data fine-tuning, applied directly to the base LLM, inherently exerted a higher behavioral priority than the agent techniques and prompting strategies co-designed to instruct the model. Consequently, when behavioral cues conflicted, the model would preferentially adhere to patterns learned from the fine-tuning data.

The virtual patient experiment utilized 300 unused patient profiles to evaluate clinical decision impacts (history-taking, diagnosis and test ordering). Referral reports from both variants were blindly evaluated by the same expert panels as in the PreA development, using validated five-point Likert scales for completeness, appropriateness, and clinical relevance (Supplementary Table 5). Inter-rater reliability for these assessments was high ($\kappa > 0.80$), and group comparisons were conducted using the two-tailed nonparametric Mann–Whitney $U$-tests.

### Randomized controlled trial

In this pragmatic, multicenter RCT, patients were randomized to use PreA independently (PreA-only), with staff support (PreA-human) or not use it (No-PreA) before specialist consultation. The PreA-human arm was included to assess PreA's autonomous capacity. The primary comparison was between the PreA-only and No-PreA arms, with a secondary comparison between PreA-only and PreA-human arms. The trial's primary end points were to evaluate the effectiveness of the PreA in enhancing operational efficiency and patient-centered care delivery in high-volume hospital settings, as measured by consultation duration, care coordination, and ease of communication. The PreA chatbot used in the RCT was frozen before patient enrolment. Examples of patient interaction with PreA are provided in Supplementary Tables 6–8.

**Participants.** Participants must demonstrate a need for health consultation or express a willingness to engage in PreA health consultations. Other inclusion criteria were (1) aged between 20 years and 80 years; (2) visit the participating physicians at the study medical centers; (3) eligible for communicative interaction via mobile phone; (4) eligible to complete the post-consultation questionnaires; and (5) have signed informed consent. Exclusion criteria were (1) the presence of psychological disorders; (2) any other medical events that are determined ineligible for LLM-based conversation; and (3) refusal to sign informed consent. No co-design stakeholders participated in the RCT.

**Intervention and comparators.** Participants were randomly assigned to one of three study arms: (1) the PreA-only group, independently interacting with the PreA via mobile phone before their physician consultation; (2) the PreA-human group, interacting with the PreA under the guidance of a medical assistant; and (3) the control group, receiving standard physician-only care (No-PreA). In the PreA-human group, participants were informed that PreA's interface was similar to WeChat, which has been used by hospitals for patient portal registries and hospital visit payments, and were offered technical support. Participants in the PreA-only arm used the tool independently without assistance.

For both PreA-assisted arms, a PreA-generated referral report was provided to specialist physicians for review via the patient's mobile phone before any face-to-face interaction. This design was implemented to prevent direct copying of content into clinical notes. Physicians were requested to rate the report's value for facilitating care. In the No-PreA control arm, physicians reviewed routine reports containing only patient sex and age.

Following consultations, patient and care partner experiences were captured via a post-consultation questionnaire (Supplementary Table 1). Physicians provided feedback at the end of their shifts (Supplementary Table 2).

**Outcomes.** The primary outcomes were consultation duration, physician-rated care coordination, and patient-rated ease of communication. These metrics were selected based on co-design feedback, which identified time efficiency and care coordination as critical for adoption in high-workload settings, and are established proxies for clinical effectiveness and patient-centered care[38].

Secondary outcomes included physician workload (measured as patients seen per shift and compared between participating and matched nonparticipating physicians); patient-reported experiences of physician attentiveness, satisfaction, interpersonal regard and future acceptability; physician-reported assessments of PreA's utility, ease of communication, and workload relief; and clinical decision-making patterns, derived from a quantitative analysis of clinical notes.

Consultation duration, patient volume and clinical notes were extracted from the PreA platform and hospital electronic records. Physician-perceived care coordination was measured via a five-point Likert scale rating the helpfulness of the PreA report in facilitating care. Patient-reported and other physician-reported outcomes were collected using prespecified, five-point Likert scale questionnaires (Supplementary Tables 1 and 2). These instruments demonstrated robust internal consistency (Cronbach's $\alpha > 0.80$ for all domains) and face validity, established through iterative feedback from 20 laypersons and five clinicians to ensure relevance to outpatient contexts. Clinical notes were extracted from all three trial arms and included five core clinical reasoning domains: history-taking (chief complaint, history of present illness and past medical history), physical examination, diagnosis, test ordering and treatment plans.

**Sample size.** The sample size was calculated for the primary comparison between the PreA-only and No-PreA arms. The target minimum sample size of 2,010 participants (670 per study arm) was prespecified based on a power analysis using preliminary data from the pilot study of 90 patients. This minimum target sample size ensured sufficient power (>80%) for the primary outcome at a significance level of 0.05.

**Recruitment.** The clinical research team approached adult patients from waiting rooms who were scheduled to see the participating physicians. For pediatric patients or adults without a mobile device, their caregivers were contacted. Interested individuals received a comprehensive study description, which emphasized the exploratory nature of the research and clarified that any advice rendered by PreA serves solely as a reference and should not be utilized as a definitive basis for disease therapy. After providing informed consent and having their questions addressed, eligible individuals who met the inclusion and exclusion criteria were formally enrolled. Recruitment was conducted from 8 Feb to 30 April 2025.

**Randomization and blinding.** Participants were allocated to one of the three groups using an individual-level, computer-generated randomization sequence without stratification. Allocation was concealed to prevent selection bias. This trial was single-blinded: while the patients knew their group assignments (PreA-only, PreA-human, or No-PreA), the physicians were uninformed about the PreA-intervention groups (PreA-only or PreA-human). Furthermore, research staff involved in data analysis remained blinded to group assignments throughout the study.

**Statistical analysis.** *Analysis of healthcare delivery.* We assessed baseline covariate balance across the three groups using an ANOVA for continuous variables and a chi-squared test for categorical variables.

For intergroup comparisons, we evaluated the distribution of scale values; two-sample Student's $t$-tests with unequal variances were used for approximately normal data, while a nonparametric Mann–Whitney $U$-test was applied to skewed distributions. All tests were two-tailed with a significance threshold of $P < 0.05$. The Benjamini–Hochberg procedure was applied to correct for multiple comparisons. The relative treatment effect for the primary comparison (PreA-only versus No-PreA) was calculated as the difference in means divided by the mean of the No-PreA group. Secondary analyses evaluated the consistency of findings across demographic and socioeconomic subgroups. Python v.3.7 and R v.4.3.0 were used to perform the statistical analyses and present the results.

We conducted a matched-pairs analysis to assess the impact of PreA on physician workload. Matching criteria included medical specialty, working shift, age group (≤45 years and >45 years), sex, and professional title (chief, associate chief and attending). The outcome is the number of patients seen per clinical shift. A second matched-pairs analysis was conducted to assess the effect on patient waiting times. For this analysis, we matched participating physicians to a distinct control group on the same covariates (replacing working shift with working week to accommodate the matching on patient volume) and the number of patients seen per shift. For both analyses, statistical significance between participating physicians and their matched controls was assessed using two-sided Wilcoxon signed-rank tests to account for matched data.

*Analysis of clinical notes.* We performed a prespecified classification analysis to detect systematic differences between clinical notes from PreA-assisted arms (PreA-only and PreA-human groups) and the No-PreA arm. A randomly selected subset of notes (PreA-only, $n = 291$; PreA-human, $n = 285$; No-PreA, $n = 300$) was retrieved in compliance with hospital data privacy rules. Notes were partitioned into training and test sets (2:1 ratio). A binary classifier was trained to distinguish PreA-assisted from No-PreA notes, with classification performance evaluated using the F1 score, defined as the harmonic mean of precision and recall F1 Score=TP/(2TP + FP + FN), where TP, FP and FN denote true positives, false positives and false negatives, respectively. Under the null hypothesis of no intergroup differences, classifier performance would be random. A statistically significant F1 score exceeding this baseline ($\Delta$F1 > 0.02) would indicate distinguishable clinical decision-making patterns attributable to PreA-assisted notes.

The classifier was trained in two stages. First, a Med-BERT encoder generated contextualized embeddings of the clinical notes[55]. Second, a binary classification layer was trained on these embeddings using supervised SimCSE[56], a contrastive learning approach that minimized embedding distance within the PreA-assisted group while maximizing the distance to the No-PreA group. Statistical significance was assessed with one-sided bootstrap tests (1,000 samples).

A prespecified domain-specific analysis further compared clinical decision-making across five domains. For unstructured text (history-taking and physical examination), Med-BERT embeddings were generated and projected into a two-dimensional latent space (UMAP1 and UMAP2) via Uniform Manifold Approximation and Projection for comparative distribution analysis. For structured non-normal count data (number of diagnoses, number of tests ordered and number of treatments), documented counts were compared between groups using nonparametric Mann–Whitney $U$-tests.

*Comparison of referral report with clinical notes.* A post hoc analysis evaluated the concordance and quality of PreA-generated referral reports versus physician-authored clinical notes. Agreement was defined as substantial alignment (exact match, near-identical content or inclusion of accepted differential diagnoses). The same expert panel from the comparative simulation studies (two board-certified primary care physicians and two senior residents) performed blinded ratings of report and note quality on a five-point Likert scale for completeness, appropriateness and clinical relevance (Supplementary Table 3). They assessed three domains relevant to primary care referrals: history-taking, diagnosis and test ordering; physical examination and treatment plans were excluded. Each case was evaluated by two experts. The analysis used the same subset of patient cases as the classification analysis. Inter-rater reliability was high (κ > 0.80). Group comparisons were performed using a nonparametric Mann–Whitney $U$-test.

### Reporting summary

Further information on research design is available in the Nature Portfolio Reporting Summary linked to this article.

## Data availability

The study protocol is provided in the Supplementary Information. Source data are provided in Tables and Extended Data Tables and can be accessed via the code repository (https://github.com/ShashaHan-collab/PreA-OutpatientRCT)[57]. Raw conversation data are not publicly available due to the need to protect participant privacy, in accordance with the ethical approval for this study. Anonymized, nondialogue individual-level data underlying the results can be requested by qualified researchers for academic use. Requests should include a research proposal, statistical analysis plan and justification for data use, and can be submitted via email to S.H. (hanshasha@pumc.edu.cn). All requests will be reviewed by the Chinese Academy of Medical Sciences & Peking Union Medical College and the ethics committee of the First Affiliated Hospital of Guilin Medical University. Review of the proposals may take up to 2 months, and approved requests will be granted access via a secure platform after execution of a data access agreement.

## Code availability

Comparative statistical analyses were detailed in the paper. Code for classification analysis and data visualization can be found at https://github.com/ShashaHan-collab/PreA-OutpatientRCT (ref. 57). The PreA chatbot is not publicly available as it is the subject of ongoing commercial licensing discussions and is protected intellectual property held by the Chinese Academy of Medical Sciences & Peking Union Medical College, intended for development as a regulated medical device. To preserve commercial viability and prevent the unregulated use of a patient-facing clinical tool, public release is not permitted at this time. To support validation and collaborative academic research, the core PreA model can be made available to qualified researchers upon a formal request to S.H. (hanshasha@pumc.edu.cn), subject to a data-sharing agreement, ethical approvals and a commitment to appropriate safety protocols.

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

## Acknowledgements

S.H. is supported by the National Natural Science Foundation of China (no. 82304269), the Chinese Academy of Medical Sciences Innovation Fund for Medical Sciences (no. 2023-I2M-3-008) and the Key Research and Development Program of Jiangxi Province (no. 20244AFI92004). L.M. is supported by the National Natural Science Foundation of China (no. 82260008) and the 2023 Bagui Young Top Talents Training Project of Guangxi Zhuang Autonomous Region. L.Z. is supported by the National Natural Science Foundation of China (no.62532001). Y.L. is supported by the Fundamental Research Funds for the Central Universities (no. 310425209503). The funders had no role in study design and conduct of the study; collection, management, analysis and interpretation of the data; preparation, review, or approval of the manuscript; and decision to submit the manuscript for publication. We appreciate the collaboration of the physicians, attending physicians, nurses, and administrative staff at the First Affiliated Hospital of Guilin Medical University and the Affiliated Hospital of Gansu Medical College during both the co-design phase and the implementation of the RCT.

## Author contributions

S.H. conceived of the research idea. All authors contributed to the study design. J. Zha., J. Zho., L.Z., L.M. and S.H. supervised the study. X.T., S.Z., K.D., S.L., Y.L., B.W., Q.H., W.C., M.S., E.M, X.C., H.H. and S.H. carried out model development, model validation, data acquisition, data curation, data analysis and data visualization. S.H. prepared the first draft of the paper. All authors critically reviewed the manuscript and approved the final submission.

## Competing interests

J. Zha. and J. Zho. are employees of WeChat AI, Tencent, which provided computational resources and technical support for this research. The other authors declare no competing interests.

## Additional information

**Extended data** is available for this paper at https://doi.org/10.1038/s41591-025-04176-7.

**Correspondence and requests for materials** should be addressed to Libing Ma or Shasha Han.

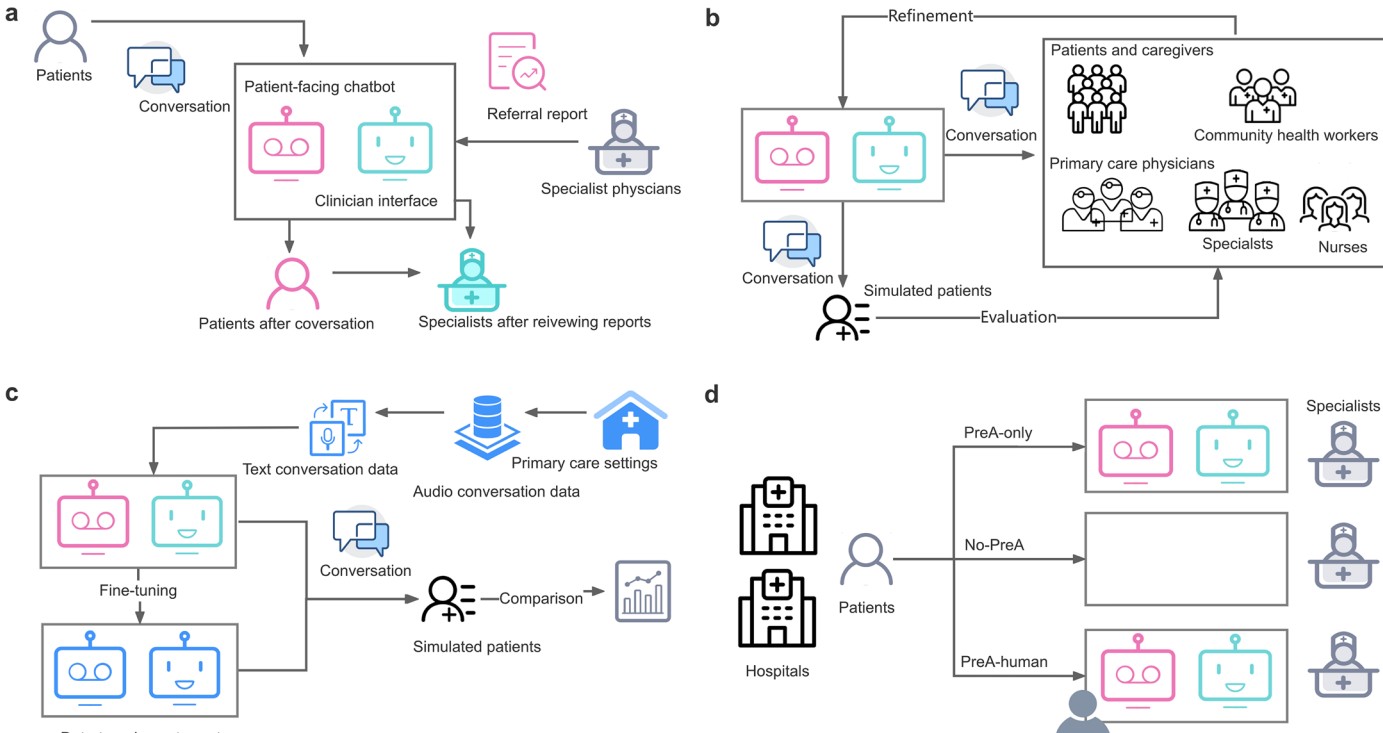

**Peer review information** *Nature Medicine* thanks Guosheng Yin and the other, anonymous, reviewer(s) for their contribution to the peer review of this work. Primary Handling Editor: Lorenzo Righetto, in collaboration with the *Nature Medicine* editorial team. Peer reviewer reports are available.

**Extended Data Fig. 1 | Study design and workflow. a**, The co-designed architecture and clinical integration of PreA. The system comprises a patient-facing chatbot and a clinician interface. Patients first interact with the chatbot, which generates a structured referral report for the specialist to review via the clinician interface prior to standard consultation. **b**, The two-cycle co-refinement process. The first cycle involved adversarial testing with community and clinical stakeholders. The second cycle used GPT-4-powered virtual patient simulations to optimize the model against co-designed evaluation metrics. **c**, Experimental comparison of the co-designed PreA model against a local data-tuned counterpart, created by fine-tuning the base PreA model on local primary care dialogs. **d**, Multicenter randomized controlled trial design. Patients were randomized to one of three arms before specialist consultation: PreA-only (independent use of PreA), PreA-human (staff-supported use of PreA), or a No-PreA control (usual care).

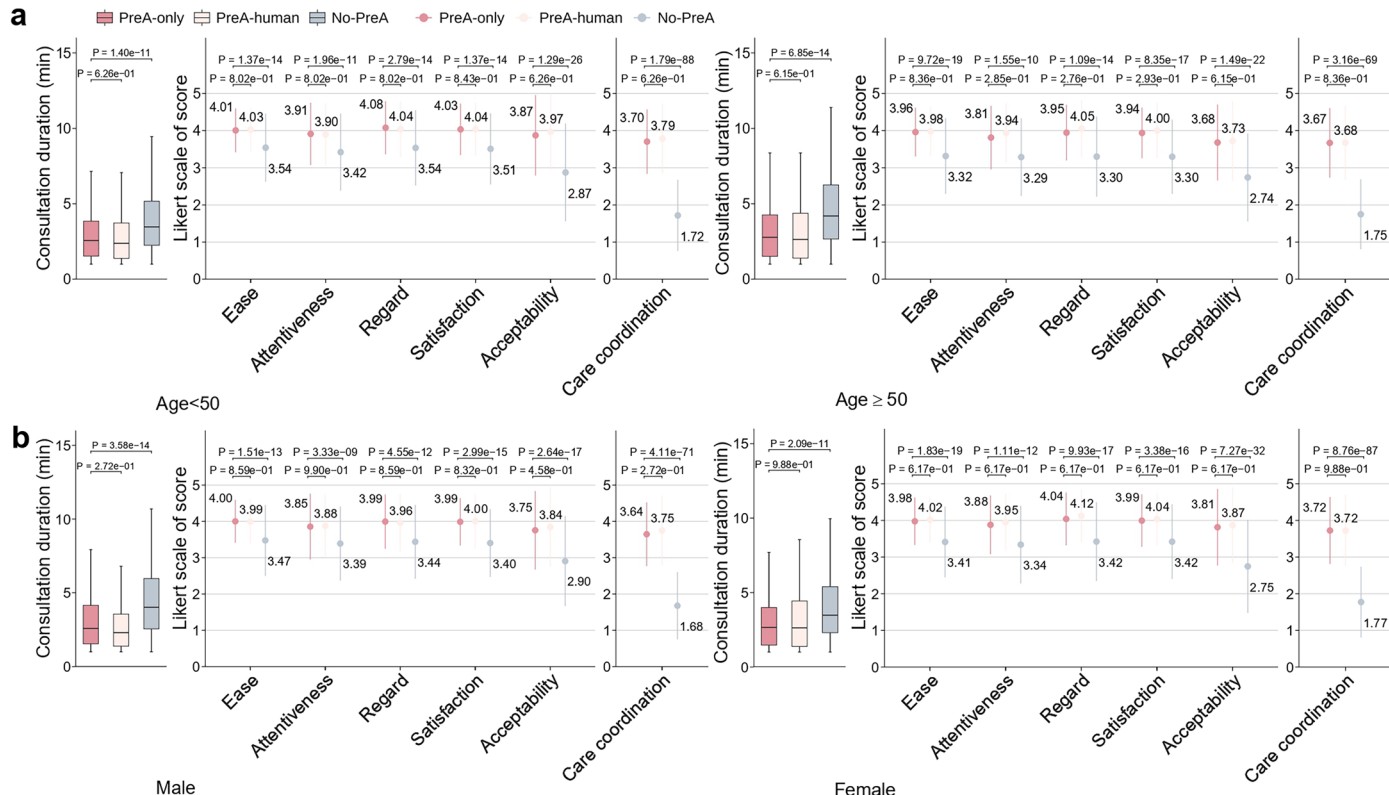

**Extended Data Fig. 2 | Consultation duration and experience stratified by age and sex. a**, By age group. **b**, By sex. Box plots depict patient consultation time across the three trial arms (PreA-only, PreA-human, No-PreA). The center line indicates the median, the box boundaries the first and third quartiles, and the whiskers extend to the most extreme data points within 1.5 × IQR. Dot plots show patient-reported experience metrics (ease of communication, perceived physician attentiveness, interpersonal regard, patient satisfaction, and future

acceptability) and physician-reported perceived value on care coordination, with error bars representing standard deviation. Sample sizes for each subgroup are provided in Table 1. We assessed the normality of value distributions and used two-sample t-tests with unequal variances for intergroup comparisons. For significantly skewed dimensions, we employed non-parametric Mann-Whitney *U*-tests. All tests were two-tailed. The Benjamini-Hochberg adjustment was applied for multiple testing corrections based on the total number of tests.

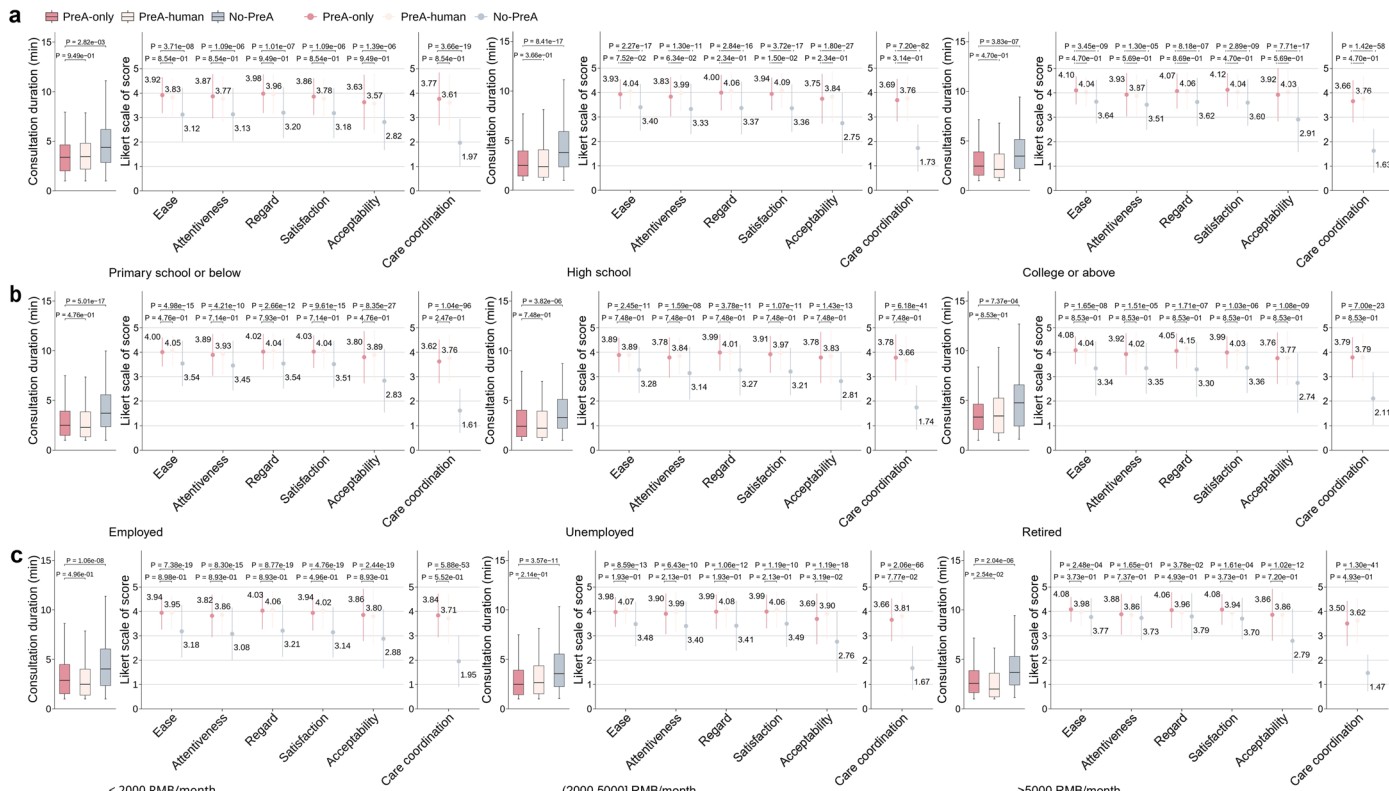

**Extended Data Fig. 3 | Consultation duration and experience stratified by socioeconomic status. a**, By education. **b**, By work status. **c**, By income level. Box plots depict patient consultation time across the three trial arms (PreA-only, PreA-human, No-PreA). The center line indicates the median, the box boundaries the first and third quartiles, and the whiskers extend to the most extreme data points within 1.5 × IQR. Dot plots show patient-reported experience metrics (ease of communication, perceived physician attentiveness, interpersonal regard, patient satisfaction, and future acceptability) and physician-reported perceived value on care coordination, with error bars representing standard deviation. Sample sizes for each subgroup are provided in Table 1. We assessed the normality of value distributions and used two-sample t-tests with unequal variances for intergroup comparisons. For significantly skewed dimensions, we employed non-parametric Mann-Whitney U-tests. All tests were two-tailed. The Benjamini-Hochberg adjustment was applied for multiple testing corrections based on the total number of tests.

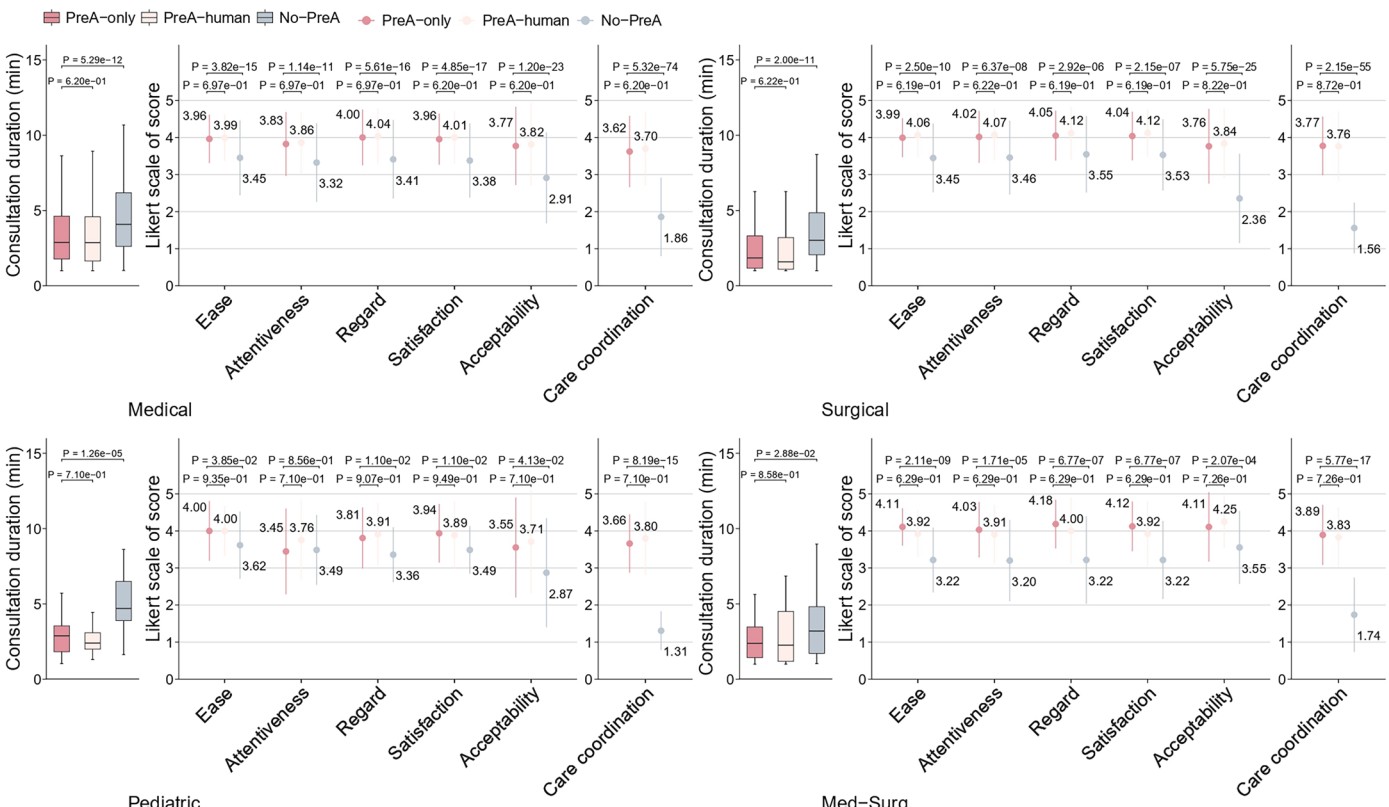

**Extended Data Fig. 4 | Consultation duration and experience stratified by medical disciplines.** Box plots depict patient consultation time across the three trial arms (PreA-only, PreA-human, No-PreA). The center line indicates the median, the box boundaries the first and third quartiles, and the whiskers extend to the most extreme data points within 1.5 × IQR. Dot plots show patient-reported experience metrics (ease of communication, perceived physician attentiveness, interpersonal regard, patient satisfaction, and future acceptability) and

physician-reported perceived value on care coordination, with error bars representing standard deviation. Sample sizes for each subgroup are provided in Table 1. We assessed the normality of value distributions and used two-sample t-tests with unequal variances for intergroup comparisons. For significantly skewed dimensions, we employed non-parametric Mann-Whitney U-tests. All tests were two-tailed. The Benjamini-Hochberg adjustment was applied for multiple testing corrections based on the total number of tests.

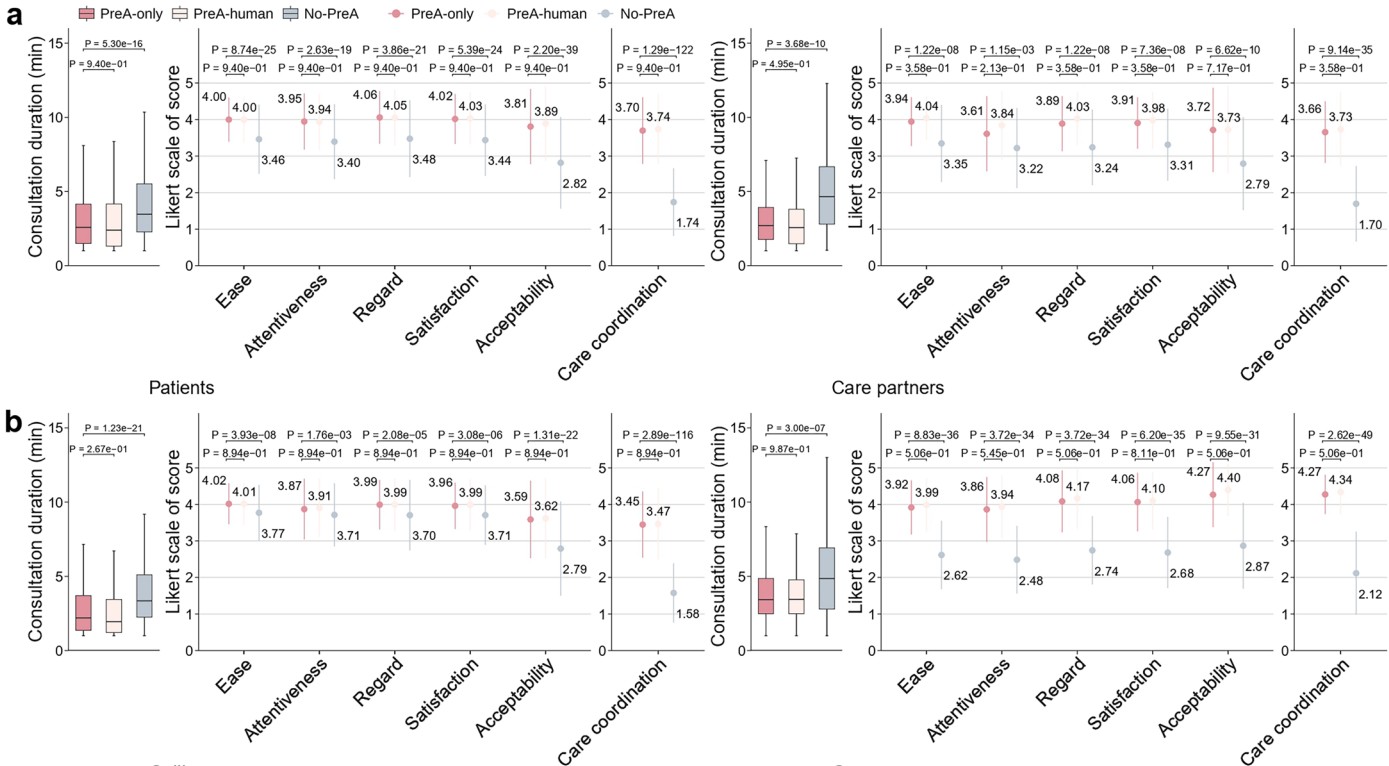

**Extended Data Fig. 5 | Consultation duration and experience stratified by participant types and study settings. a**, By participant types. **b**, study settings. Box plots depict patient consultation time across the three trial arms (PreA-only, PreA-human, No-PreA). The center line indicates the median, the box boundaries the first and third quartiles, and the whiskers extend to the most extreme data points within 1.5 × IQR. Dot plots show patient-reported experience metrics (ease of communication, perceived physician attentiveness, interpersonal regard, patient satisfaction, and future acceptability) and physician-reported perceived value on care coordination, with error bars representing standard deviation. Sample sizes for each subgroup are provided in Table 1. We assessed the normality of value distributions and used two-sample t-tests with unequal variances for intergroup comparisons. For significantly skewed dimensions, we employed non-parametric Mann-Whitney $U$-tests. All tests were two-tailed. The Benjamini-Hochberg adjustment was applied for multiple testing corrections based on the total number of tests.

**Extended Data Table 1 | Distributions of medical departments across three trial groups**

| Category, No. | Medical departments[a] | Total 2,069 | PreA-only 691 | PreA-human 689 | No-PreA 689 |
|---|---|---|---|---|---|
| Medical,1181 | Cardiology | 202 | 64 | 70 | 68 |
| | Endocrinology | 179 | 62 | 57 | 60 |
| | Gastroenterology | 184 | 69 | 47 | 68 |
| | Geriatrics | 11 | 5 | 4 | 2 |
| | Gynecology (Medicine) | 29 | 12 | 9 | 8 |
| | Hematology | 14 | 5 | 3 | 6 |
| | Nephrology | 37 | 13 | 12 | 12 |
| | Neurology | 194 | 69 | 68 | 57 |
| | Respiratory Medicine | 181 | 56 | 59 | 66 |
| | Physical Medicine and Rehab | 9 | 2 | 4 | 3 |
| | Rheumatology and Immunology | 25 | 7 | 11 | 7 |
| | Traditional Chinese Medicine | 121 | 37 | 50 | 34 |
| Surgical, 559 | Gastrointestinal Surgery | 76 | 22 | 29 | 25 |
| | Hepatobiliary Surgery | 25 | 8 | 7 | 10 |
| | Neurosurgery | 18 | 5 | 10 | 3 |
| | Orthopedics | 129 | 45 | 43 | 41 |
| | ENT (Otorhinolaryngology) | 44 | 21 | 14 | 9 |
| | Thoracic Surgery | 16 | 4 | 8 | 4 |
| | Thyroid and Breast Surgery | 204 | 60 | 56 | 88 |
| | Urology | 46 | 13 | 19 | 14 |
| Med-Surg[b], 194 | Dermatology | 137 | 43 | 44 | 50 |
| | Ophthalmology | 28 | 13 | 7 | 8 |
| | Stomatology | 29 | 9 | 13 | 7 |
| Pediatric,131 | Pediatrics | 131 | 47 | 45 | 39 |

Notes: **a**. The distribution of medical disciplines among the three intervention groups was balanced, with a P value of 0.497 from the Chi-squared test. **b**. Med-Surg represents the department that provide both medical and surgial interventions for patients.

**Extended Data Table 2 | Consultation workflow, physician ratings, and patient experience in the RCT**

| | PreA-only | PreA-human | No-PreA | P value PreA-only vs. No-PreA | P value PreA-only vs. PreA-human | Relative difference of PreA-only compared to No-PreA |
|---|---|---|---|---|---|---|
| **Clinical workflow** | | | | | | |
| Consultation duration | 3.14 ± 2.25 | 3.17 ± 2.87 | 4.41 ± 2.77 | 7.82e-24 | 0.17 | -28.7% (22.7–34.8) |
| **Physician experience** | | | | | | |
| Care coordination | 3.69 ±0.90 | 3.74 ±0.97 | 1.73 ±0.95 | 1.46e-157 | 0.22 | +113.1% (107.4–118.7) |
| **Patient experience** | | | | | | |
| Ease of communication | 3.99 ±0.62 | 4.01 ±0.62 | 3.44 ±0.97 | 4.06e-32 | 0.67 | +16.0% (13.5–18.5) |
| Physician attentiveness | 3.87 ±0.85 | 3.92 ±0.81 | 3.36 ±1.04 | 1.74e-20 | 0.40 | +15.1% (12.1–18.1) |
| Interpersonal regard | 4.02 ±0.73 | 4.05 ±0.76 | 3.43 ±1.05 | 1.48e-27 | 0.39 | +17.2% (14.4–20.0) |
| Patient satisfaction | 3.99 ±0.69 | 4.02 ±0.73 | 3.41 ±0.98 | 3.71e-30 | 0.26 | +17.0% (14.3–19.6) |
| Future acceptability | 3.79 ±1.06 | 3.86 ±1.06 | 2.81 ±1.26 | 1.94e-48 | 0.15 | +34.7% ( 30.4–39.1) |

Notes: Values for each group were presented in mean ± standard deviation. The relative differences were presented in the mean percentage (95% CI). We assessed the normality of value distributions and used two-sample two-tailed t-tests with unequal variances for intergroup comparisons. For significantly skewed dimensions, we employed non-parametric Mann-Whitney U tests. All tests were two-tailed. The Benjamini-Hochberg adjustment was applied for multiple testing corrections based on the total number of tests.

**Extended Data Table 3 | P values of statistical comparisons on domain-specific clinical notes between groups**

| Domain of clincal notes | Recording types | PreA-only vs No-PreA | PreA-only vs PreA-human |
|---|---|---|---|
| History taking | Unstructured free-text entries | UMAP1: 0.10 | UMAP1: 0.70 |
| | | UMAP2: 0.46 | UMAP2: 0.17 |
| Physical examination | Unstructured free-text entries | UMAP1: 0.84 | UMAP1: 0.76 |
| | | UMAP2: 0.90 | UMAP2: 0.39 |
| No. of diagnosis | Structured entires | 0.10 | 0.49 |
| No. of ordered tests | Structured entires | 0.52 | 0.29 |
| No. of treatments | Structured entires | 0.48 | 0.25 |

# Reporting Summary

## Statistics

For all statistical analyses, confirm that the following items are present in the figure legend, table legend, main text, or Methods section.

| n/a | Confirmed | |
|---|---|---|
| ☐ | ☒ | The exact sample size (*n*) for each experimental group/condition, given as a discrete number and unit of measurement |
| ☐ | ☒ | A statement on whether measurements were taken from distinct samples or whether the same sample was measured repeatedly |
| ☐ | ☒ | The statistical test(s) used AND whether they are one- or two-sided<br>*Only common tests should be described solely by name; describe more complex techniques in the Methods section.* |
| ☐ | ☒ | A description of all covariates tested |
| ☐ | ☒ | A description of any assumptions or corrections, such as tests of normality and adjustment for multiple comparisons |
| ☐ | ☒ | A full description of the statistical parameters including central tendency (e.g. means) or other basic estimates (e.g. regression coefficient) AND variation (e.g. standard deviation) or associated estimates of uncertainty (e.g. confidence intervals) |
| ☐ | ☒ | For null hypothesis testing, the test statistic (e.g. *F*, *t*, *r*) with confidence intervals, effect sizes, degrees of freedom and *P* value noted<br>*Give P values as exact values whenever suitable.* |
| ☒ | ☐ | For Bayesian analysis, information on the choice of priors and Markov chain Monte Carlo settings |
| ☒ | ☐ | For hierarchical and complex designs, identification of the appropriate level for tests and full reporting of outcomes |
| ☒ | ☐ | Estimates of effect sizes (e.g. Cohen's *d*, Pearson's *r*), indicating how they were calculated |

*Our web collection on statistics for biologists contains articles on many of the points above.*

## Software and code

Policy information about availability of computer code

| Data collection | Our co-designed GPT-4-powered PreA platform (OpenAI; GPT-4o mini) chatbot was used to collect coversational and survey data. |
|---|---|
| Data analysis | Comparative statistical analyses were detailed in the paper. Python 3.7 and R 4.3.0 were used to perform the statistical analyses and present the results. Code for classification analysis and data visualization can be found at the following link (https://github.com/ShashaHan-collab/PreA-OutpatientRCT). |

For manuscripts utilizing custom algorithms or software that are central to the research but not yet described in published literature, software must be made available to editors and reviewers. We strongly encourage code deposition in a community repository (e.g. GitHub). See the Nature Portfolio guidelines for submitting code & software for further information.

## Data

Policy information about availability of data

All manuscripts must include a data availability statement. This statement should provide the following information, where applicable:
- Accession codes, unique identifiers, or web links for publicly available datasets
- A description of any restrictions on data availability
- For clinical datasets or third party data, please ensure that the statement adheres to our policy

The study protocol is provided in the Supplementary Information. Source data are provided in Tables and Extended Data Tables and can be accessed via the code repository (https://github.com/ShashaHan-collab/PreA-OutpatientRCT). Raw conversation data are not publicly available due to the need to protect participant

# Research involving human participants, their data, or biological material

Policy information about studies with [human participants or human data](). See also policy information about [sex, gender (identity/presentation), and sexual orientation]() and [race, ethnicity and racism]().

| | |
|---|---|
| Reporting on sex and gender | Our trial analysis included 2,069 participants (691 with PreA-only, 689 with PreA-human, and 689 with No-PreA), with a mean age of 47.6 (SD 14.6), and 1,141 women (55.1%) and 928 men (44.9%). |
| Reporting on race, ethnicity, or other socially relevant groupings | The majority of participants (1,620, 78.3%) were patients themselves, while the remainder were the patients' care partners. Less than half (881, 42.6%) of participants were currently unemployed or retired, and 770 (37.2%) of participants had an income (including income from an office, employment on a full-time, part-time, or casual basis, or a pension from former employment) of less than 2000 RMB per month. 313 (15.1%) participants had education attainment less than primary school or below, 1073 (51.9 %) participants had high school, and 683 (33.0%) had college or above.<br><br>First, the generalizability of our time-reduction findings may be context-dependent, as our study was conducted in high-volume, resource-constrained hospital settings. The effectiveness of PreA is intrinsically tied to this environment of high clinical demand and standardized workflows, and validation in diverse healthcare systems is warranted. Second, the single-blinded, pragmatic trial design, while reflecting real-world conditions where patients would naturally know their pre-consultation experience, introduces potential performance bias as patients were aware of their group assignment. However, several factors mitigate this concern: the concordance of findings across multiple outcome assessments, the absence of significant differences in clinical documentation between groups, and the alignment of control group consultation times with established practice patterns. |
| Population characteristics | Our trial analysis included 2,069 participants (691 with PreA-only, 689 with PreA-human, and 689 with No-PreA), with a mean age of 47.6 (SD 14.6), and 1,141 women (55.1%) and 928 men (44.9%). The majority of participants (1,620, 78.3%) were patients themselves, while the remainder were the patients' care partners. Less than half (881, 42.6%) of participants were currently unemployed or retired, and 770 (37.2%) of participants had an income (including income from an office, employment on a full-time, part-time, or casual basis, or a pension from former employment) of less than 2000 RMB per month. 313 (15.1%) participants had education attainment less than primary school or below, 1073 (51.9 %) participants had high school, and 683 (33.0%) had college or above. 1186 (57.3%) of participants consulted for medical specialty, 558 (27.0%) for surgical, 194 (9.4%) for a specialty that provides both medical and surgical treatments, and the remaining consulted for pediatrics. These baseline covariates were well-balanced across the three intervention groups, with no statistically significant differences in covariate distributions (Table 1). |
| Recruitment | In the trial, we recruited 111 physicians across 24 medical disciplines (Supplementary Table 7) from the two medical centers. The clinical research team proactively contacted potential adult patients from the waiting room who were scheduled to see the participating physicians. For pediatric patients and adult patients who did not have a smartphone, the clinical research team contacted their caregivers. For those who indicated interest, the research team provided comprehensive descriptions of the study, emphasizing that it is exploratory and that any advice rendered by PreA serves solely as a reference and should not be utilized as a definitive basis for disease therapy. Patients and caregivers received an informed consent form before enrollment and will have the opportunity to ask questions. After this process, potential patients and caregivers who met the established inclusion and exclusion criteria were formally recruited. |
| Ethics oversight | The Chinese Academy of Medical Sciences & Peking Union Medical Colleges and the local medical ethics committee of the First Affiliated Hospital of Guilin Medical University approved the study. The institutional review boards of the Affiliated Hospital of Gansu Medical College approved the study protocol based on their review and the approval from the medical ethics committee of the First Affiliated Hospital of Guilin Medical University. The trial followed the Declaration of Helsinki and the International Conference of Harmonization Guidelines for Good Clinical Practice. We obtained informed consent from all participants (physicians, patients, and caregivers) in this study. All participants were informed that this was an exploratory experiment, and the results should not be interpreted as direct guidance for clinical interventions at this stage. This study implemented stringent data protection measures, ensuring that all data were anonymized and encrypted to protect privacy. This trial is registered at the Chinese Clinical Trial Registry (identifier: ChiCTR2400094159). The trial protocol and statistical analysis plan were provided in the Supplementary Materials. |

Note that full information on the approval of the study protocol must also be provided in the manuscript.

# Field-specific reporting

Please select the one below that is the best fit for your research. If you are not sure, read the appropriate sections before making your selection.

☒ Life sciences ☐ Behavioural & social sciences ☐ Ecological, evolutionary & environmental sciences

For a reference copy of the document with all sections, see [nature.com/documents/nr-reporting-summary-flat.pdf](nature.com/documents/nr-reporting-summary-flat.pdf)

# Life sciences study design

All studies must disclose on these points even when the disclosure is negative.

| | |
|---|---|
| Sample size | The sample size was estimated to depend on the primary comparison, PreA-only versus No-PreA. The target minimum sample size of 2010 participants (670 participants per study arm) was prespecified based on a power analysis using the preliminary data of 90 patients in the pilot study before study enrollment. This minimum target sample size ensured sufficient power (>80%) for the primary outcome at a significance level of 0.05. |
| Data exclusions | 2,332 patients and their care partners were evaluated for eligibility, with 194 either opting out or being excluded for various reasons (Fig. 2). This left 2,138 patients who were randomly assigned to the PreA-only group (n= 712), PreA-human group (n= 713), or the No PreA group (n= 713) using sealed envelopes for a straightforward 1:1:1 allocation. Of these, 69 patients later chose to opt-out or were removed for different reasons. |
| Replication | Stratified analyses demonstrated consistent reductions in physician consultation duration. Notably, these reductions were observed across age groups, sex, educational attainment, work status, income levels, medical disciplines (medical medicine, surgery, mix of medical medicine and surgery, pediatrics), study sites (Guilin/Gansu) and participant type (patients/care partners), with PreA-only showing significant reductions compared to No-PreA, and no significant differences compared to PreA-human (Supplementary Figs. 1-4). |
| Randomization | We used individual-level parallel randomization without stratification, utilizing a computer-generated random sequence for participant assignment to each experimental group. |
| Blinding | This trial was single-blinded: While the patients knew their group assignments, the physicians were uninformed about the PreA-intervention groups (PreA-only or PreA-human), and the researchers were also unaware of the assignments. |

# Reporting for specific materials, systems and methods

We require information from authors about some types of materials, experimental systems and methods used in many studies. Here, indicate whether each material, system or method listed is relevant to your study. If you are not sure if a list item applies to your research, read the appropriate section before selecting a response.

## Materials & experimental systems

| n/a | Involved in the study |
|---|---|
| ☒ | ☐ Antibodies |
| ☒ | ☐ Eukaryotic cell lines |
| ☒ | ☐ Palaeontology and archaeology |
| ☒ | ☐ Animals and other organisms |
| ☐ | ☒ Clinical data |
| ☒ | ☐ Dual use research of concern |
| ☒ | ☐ Plants |

## Methods

| n/a | Involved in the study |
|---|---|
| ☒ | ☐ ChIP-seq |
| ☒ | ☐ Flow cytometry |
| ☒ | ☐ MRI-based neuroimaging |

## Clinical data

Policy information about clinical studies

All manuscripts should comply with the ICMJE guidelines for publication of clinical research and a completed CONSORT checklist must be included with all submissions.

| | |
|---|---|
| Clinical trial registration | Chinese Clinical Trial Registry identifier: ChiCTR2400094159 |
| Study protocol | The study protocol is provided in Supplemetary Information. |
| Data collection | To evaluate the impact of PreA in a real-world setting, we conducted a multicenter, parallel-group RCT across 24 medical disciplines at two tertiary care centers in western China: the First Affiliated Hospital of Guilin Medical University and the Affiliated Hospital of Gansu Medical College, from February 8, 2025, to April 30, 2025. |
| Outcomes | The primary outcomes were the duration of physician-patient consultations, physician perception of primary-secondary care coordination, and patient perception of ease of communication (assessed using validated questionnaires). Secondary outcomes included: (1) physician workload (measured by number of patients per shift); (2) patient perceived physician attentiveness during visits, patient satisfaction, interpersonal regard, and future acceptability; (3) physician experience with perceived utility of PreA, ease of communication ease with patients, and relief of workload; (4) physician documentation practices (analyzing clinical note content). |

# Plants

Seed stocks

*Report on the source of all seed stocks or other plant material used. If applicable, state the seed stock centre and catalogue number. If plant specimens were collected from the field, describe the collection location, date and sampling procedures.*

Novel plant genotypes

*Describe the methods by which all novel plant genotypes were produced. This includes those generated by transgenic approaches, gene editing, chemical/radiation-based mutagenesis and hybridization. For transgenic lines, describe the transformation method, the number of independent lines analyzed and the generation upon which experiments were performed. For gene-edited lines, describe the editor used, the endogenous sequence targeted for editing, the targeting guide RNA sequence (if applicable) and how the editor was applied.*

Authentication

*Describe any authentication procedures for each seed stock used or novel genotype generated. Describe any experiments used to assess the effect of a mutation and, where applicable, how potential secondary effects (e.g. second site T-DNA insertions, mosiacism, off-target gene editing) were examined.*

