## [Peer Review File · Nature Medicine]

An LLM chatbot to facilitate primary-to-specialist care transitions: a randomized controlled trial

Corresponding Author: Dr Shasha Han

Version 1:

Reviewer comments:

Reviewer #1

(Remarks to the Author)

Remarks to the author:

In this study, the authors trained a co-designed pre-assessment LLM for medical referral. To assess the importance of a multi-stakeholder co-design approach to LLM training, they evaluated PreA against a counterpart model fine-tuned on local dialogues. To evaluate PreA's real-world clinical utility, the authors conducted a RCT, demonstrating that PreA enhances the patient experience, particularly in resource-limited settings or socio-economic status, and exhibits robust performance when operating independently.

Major concerns:

1. There was misalignment between the declared "referral" function of PreA and RCT study design. Patients were recruited from those who had already booked specialist consultations, and PreA was used merely as a pre-consultation step. This raises questions about its practical utility. First of all, PreA in this trial design appears to add redundancy rather than integrated into and streamline the real-world clinical workflows. Secondly, the clinical goal of this pre-consultation step was unclear. Is the goal to provide patients with potential differential diagnoses to discuss with specialists? If so, this could bias or unmask the patient-physician interactions.
2. In the RCT, as patients were recruited from the consultation waiting area, how can the researchers ensure that participants had not used commercialized LLMs (such as DeepSeek) or web references to search for information about their diseases so as to avoid bias?
3. The authors emphasize PreA's "independent operation" (PreA-only vs. PreA-human groups showed no difference), yet also advocate for co-design to mitigate risks. If PreA works independently, why is stakeholder input critical? In addition, The co-design involved multiple stakeholders, yet the RCT only tested PreA in patient-specialist interactions. This mismatch invites skepticism about why were these stakeholders included in co-design? Will their involvement lead to potential bias or influence of the model performance?
4. The authors compare co-designed PreA with a version fine-tuned on local dialogues but omit comparison to a high-quality dataset (for example, public dataset such as MedQA, or curated private dataset from urban tertiary care centers, or curated high-quality dataset from their research centers). This may bias their conclusion that "co-design works better than local finetuned" as the observed difference may be due to unfair comparison between the in simply-curated low-quality local data and additional finer curation by the 209 multi-stakeholders.
5. This manuscript contains numerous analysis results, but the writing style requires revision to enhance clarity regarding the RCT by adhering to the CONSORT-AI checklist. The results section should explicitly specify primary outcomes, secondary outcomes, exploratory outcomes, and post-deployment analyses, along with clear identification of the participant groups from which these results were derived. The current presentation of results is ambiguous, leading to confusion about whether specific results were evaluated using masked participants/evaluators or unmasked participants.
6. In the Methods section, the authors declare that "the quality of PreA reports and physical notes was rated by a panel of experts regarding completeness, appropriateness, and clinical relevance" (Page 35, Lines 716–718). However, both Figure 4 and Supplementary Table 1 present only a single 5-point Likert scale—please clarify this discrepancy. Additionally, the authors conclude that higher-quality reports are "patient-centred" (Page 21, Lines 432–433), yet in the Methods section's co-design evaluation metrics, patient-centeredness is defined as "Was the AI counselor friendly and respectful towards the patient?" (Page 26, Line 528). Please explain these inconsistencies and provide the exact definitions. Otherwise, the "patient-centered" related conclusions in the Discussion part shall be revised.

Minor concerns:

1. Please re-organize the presentation structure of the manuscript. The current Results section contains substantial non-result content, such as model design descriptions. Such material—lacking data analysis—should be simplified within the Results and relocated to the Methods section.
2. The high rate of missing physician notes (23.0–31.9% presented in Figure 4) suggests documentation non-compliance or workflow disruptions for the physicians participated in the trial. This confounds the quality of the trial.
3. The statement “In this study, we introduced PreA, a co-designed LLM-based platform, to enhance the primary-to-secondary care interface” (Page 17, Line 357) at the beginning of the Discussion part requires revision. PreA was never tested in primary care settings to filter unnecessary referrals or improve referral quality or efficiency.
4. The authors may consider re-conducting the subgroup analysis based on the economic levels of different provinces for the local data-tuned model section (Extended Fig 1), rather than the current Eastern/Western China subgroups.

(Remarks on code availability)

Reviewer #3

(Remarks to the Author)

The paper examines the use of large language models (LLMs) in outpatient settings, particularly for pre-specialist consultations. The authors developed a mobile-accessible platform called PreA, designed to streamline medical consultations and generate referral reports. They employed a collaborative, multi-stakeholder co-design approach involving diverse participants. To evaluate the platform, a randomized controlled trial (RCT) involving 2,069 patients across 24 medical disciplines was conducted in two hospitals in China. Patients were randomly placed into three groups: PreA-only, PreA-human (with medical assistant support), and standard care (No-PreA). The study found a significant reduction in consultation duration for the PreA-only group compared to standard care.

Major concerns:

1. Integrating LLMs into healthcare workflows, especially for pre-consultation purposes, is not novel. Previous research has already extensively covered AI-driven solutions in healthcare management and decision support. For example, Topol (2019) discussed the integration of human and artificial intelligence in healthcare. Chen et al. (2025) highlighted the improved diagnostic capabilities using conversational LLMs. Goh et al. (2025) demonstrated physician assistance using GPT-4 in patient care tasks. The authors did not clearly differentiate their platform, PreA, from existing technologies.
2. The authors emphasize a statistically significant reduction of approximately one minute in consultation duration. However, the practical impact of this time reduction is limited, especially since consultation duration may not be the primary factor influencing clinical efficiency. Thus, this improvement does not convincingly translate into meaningful operational or clinical enhancements.
3. The study heavily relies on indirect measures such as patient satisfaction, ease of communication, and consultation length. It lacks critical direct clinical outcomes like diagnostic accuracy improvements, accurate referral rates, or subsequent health outcomes. Additionally, there is no long-term follow-up to confirm sustained benefits or detect adverse effects.
4. Although the authors acknowledge potential biases arising from using local medical dialogue datasets, they do not thoroughly investigate this critical issue. Their analysis lacks depth regarding how these biases might affect clinical decision-making or patient outcomes.

References:

Topol, Eric J. "High-performance medicine: the convergence of human and artificial intelligence." *Nature Medicine* 25.1 (2019): 44-56.

Chen, Xi, et al. "Enhancing diagnostic capability with multi-agents conversational large language models." *NPJ digital medicine* 8.1 (2025): 159.

Goh, Ethan, et al. "GPT-4 assistance for improvement of physician performance on patient care tasks: a randomized controlled trial." *Nature Medicine* (2025): 1-6.

(Remarks on code availability)

Version 2:

Reviewer comments:

Reviewer #1

(Remarks to the Author)

The authors have adequately addressed most of my initial concerns. The response is adequate and comprehensive

This paper evaluated a LLM (PreA), which was co-designed with many stakeholders, including patients, to enhance its effectiveness and the operational efficiency in high-volume specialist hospital settings. The pragmatic RCT showed that patient care and operational efficiency augmented by this LLM (PreA) was superior to usual clinical practice, and appeared to more adequately "prepare" patients for specialist consultation.

(Remarks on code availability)

Reviewer #3

(Remarks to the Author)

This is a single-blinded, pragmatic trial design, which cannot objectively assess the differences in the primary outcome. This trial was single-blinded: While the patients knew their group assignments (PreA-only, PreA-human, or No-PreA), the physicians were uninformed about the PreA-intervention groups (PreA-only or PreA-human), and the researchers were also unaware of the assignments. All evaluators were masked throughout the study.

The primary outcome was the duration of the medical consultation between patients and physicians, defined as the time elapsed from when patients started conversing with physicians to the end of the consultation. With patients knowing their group assignments, this may cause bias.

(Remarks on code availability)

Response to referees

Reviewer #1 (Remarks to the Author):

Remarks to the author:

In this study, the authors trained a co-designed pre-assessment LLM for medical referral. To assess the importance of a multi-stakeholder co-design approach to LLM training, they evaluated PreA against a counterpart model fine-tuned on local dialogues. To evaluate PreA's real-world clinical utility, the authors conducted a RCT, demonstrating that PreA enhances the patient experience, particularly in resource-limited settings or socio-economic status, and exhibits robust performance when operating independently.

Response: We appreciate the reviewer's concise summary of our work and the accurate identification of key findings regarding PreA's effectiveness in resource-limited settings with socioeconomic diversities and its robust independent performance.

Major concerns:

1. There was misalignment between the declared "referral" function of PreA and RCT study design. Patients were recruited from those who had already booked specialist consultations, and PreA was used merely as a pre-consultation step. This raises questions about its practical utility. First of all, PreA in this trial design appears to add redundancy rather than integrated into and streamline the real-world clinical workflows. Secondly, the clinical goal of this pre-consultation step was unclear. Is the goal to provide patients with potential differential diagnoses to discuss with specialists? If so, this could bias or unmask the patient-physician interactions.

Response: We thank the reviewer for their thoughtful critique of our study design. We appreciate this opportunity to clarify the study's design and the specific function of PreA within this trial.

The primary objective of the RCT was not to evaluate PreA as a de novo referral tool, but rather to assess its utility as a pre-specialist consultation tool in a real-world clinical setting, especially in resource-limited high-volume outpatient settings. We aimed to address a critical, systemic gap in resource-poor healthcare settings, where specialists in outpatients often encounter patients who self-referred to specialists, often arriving without a medical summary or referral report. This practice led to incomplete history-taking, inefficient consultations, delayed diagnoses, and increased burnout among specialist physicians and poor consultation experience among patients (Refs 12-24).

PreA addresses this by offering general medical consultation during patient waiting periods (patients' waiting time: 34.65 ± 36.92 minutes; PreA consultation duration: 3.14 ± 2.25 minutes). The clinical goals are twofold. First, PreA generates the necessary referrals for specialists to

review before starting their usual care. Second, PreA serves as a pre-consultation communication tool for the patient to enhance patient-centred communication, which is a critical component of clinical efficacy (Ong, 1995; Stewart, 2001; Ha and Longnecker, 2010; Sharkiya, 2023). In doing so, PreA effectively streamlines the specialist's workflow by reducing consultation time and improving communication quality, directly addressing the reviewer's concern about redundancy.

We fully agree that the broader integration of such a tool would be at a community level, enabling home-based pre-specialist consultations and close integration with regional or national health information systems. However, implementing such a large-scale intervention first requires demonstrating efficacy within current practice constraints. Our hospital-based trial provides the essential, foundational evidence of effectiveness within the existing patient self-referral system, laying the groundwork for future community-level studies and systemic integration.

Regarding the second point about the masking, we acknowledged that this was a single-blinded pragmatic trial: Patients knew their group assignments (PreA-only, PreA-human, or No-PreA), the physicians were uninformed about the PreA-intervention groups (PreA-only or PreA-human), and the researchers were also unaware of the assignments. All evaluators were masked throughout the study. To improve transparency and reduce cognitive anchoring risks, the potential differential diagnoses were framed as possibilities with supporting and contradictory evidence. Furthermore, our study protocol required specialists to follow their usual care procedures after reviewing the referral report. Our pre-specific analysis of physician clinical notes showed no significant differences between the PreA-assisted and No-PreA groups, providing strong evidence that the tool did not inappropriately influence or bias the physician's clinical judgment (Results: Physician clinical decision-making). Additionally, both patients and physicians in the PreA group reported a significantly better consultation experience, indicating that the integration can enhance, rather than undermine, the quality of patient-physician interactions (Results: Patient perception of patient-centeredness and physician perception of care coordination).

In this revision, we added three PreA-patient interaction dialogues in Supplementary Tables 7-9 (copied below) to help readers better understand the pre-specialist consultation. Additionally, we have updated the manuscript to address these concerns as follows:

Main, lines 118-122: *“This collaborative effort focused on developing pre-assessment (PreA), a mobile-accessible LLM-based platform (OpenAI; GPT-4.0 mini) that provides general medical consultations to patients and generates referral reports for specialist physicians, functioning independently in clinical workflow (Methods).”*

Discussion, lines 315-318: *“The clinical goals are twofold: preparing patients for specialist consultation and creating medical referrals for specialists with differentially relevant diagnoses contextualized by both supporting and contradictory evidence.”*

Discussion, lines 395-403: “Fourth, the single-blinded, pragmatic trial design, where patient-physician interactions were unmasked between the PreA-assisted groups and the No-PreA group, carries inherent risks of bias. However, our analysis of physician clinical notes revealed no significant differences between the PreA-assisted and No-PreA groups, mitigating concerns about direct adoption or over-scrutiny of LLM-generated content. Fifth, our trial successfully establishes effectiveness within current self-referral practice constraints, providing the essential foundation for subsequent community-level studies; a home-based pre-consultation workflow represents the future integration for exploration.”

References:

Ong LML, de Haes JCJM, Hoos AM, Lammes FB. Doctor-patient communication: A review of the literature. *Soc Sci Med*. 1995;40(7):903-918.

Stewart M. Towards a global definition of patient centred care: the patient should be the judge of patient centred care. *BMJ*. 2001;322(7284):444-5.

Ha JF, Longnecker N. Doctor-patient communication: a review. *Ochsner J*. 2010;10(1):38-43.

Sharkiya SH. Quality communication can improve patient-centred health outcomes among older patients: a rapid review. *BMC Health Serv Res*. 2023: 23, 886.

Supplementary Table 7 A de-identified case of PreA interacting with real-world patients: The patient missed important information until later in the dialogue that required multi-disciplinary assessment

PreA: Hello, I'm your pre-specialist consultation assistant. You can communicate with me via text or voice messages.

Patient: I've been losing a lot of hair.

PreA: May I ask how old you are? And what is your gender?

Patient: Female, 54 years old.

PreA: Thank you for the information. How long has your hair loss been going on? Have you noticed any specific pattern of hair loss, such as localised hair loss or general hair loss?

Patient: It's all over my head. I also have trouble sleeping. The hair loss started suddenly; I used to lose much less hair before.

PreA: I see. You mentioned that the hair loss has suddenly worsened and is accompanied by poor sleep. Have you experienced significant stress or emotional changes recently? Such as stress related to work, family, or life?

Patient: No, none of these.

PreA: Alright, thank you for the additional information. Have there been any particular changes in your diet, such as changes in your diet structure or nutrient intake? Or have you taken any new medications or health supplements recently?

Patient: My diet is fine. It's just that before the Chinese New Year, I had a cold that lasted a long time. Then, when I sleep at night, I keep sweating; when I get up, my whole body is wet. Also, my ears have been ringing all the time. Sometimes I think it's the sound of water boiling, so I run over to check, but the water isn't boiling yet.

PreA: Thank you for your detailed description. You mentioned symptoms of a cold, night sweats, and tinnitus. I'd like to know if you have any other health problems, such as high blood pressure, diabetes, or other chronic diseases. Also, have you had any weight changes recently, or felt unusually fatigued?

Patient: None of those. I just sit at home all the time. I feel mentally tired, and sometimes extremely exhausted.

PreA: Thank you so much for your thoughtful responses. To sum up, it looks like you've been experiencing the following symptoms lately. Please don't hesitate to share these details openly with your specialist so they can best assist you.

Sudden worsening of general hair loss, poor sleep and night sweats, tinnitus, mental fatigue, and physical weakness.

Based on this information, I consider the following possible diagnoses:

Thyroid dysfunction (such as hyperthyroidism or hypothyroidism)

Supporting Evidence: Hair loss, fatigue, and excessive sweating can all be related to thyroid issues.

Opposing Evidence: Further tests are needed to confirm thyroid function.

Menopause-related symptoms

Supporting Evidence: Your age (54 years old) and night sweats may be related to hormonal changes.

Opposing Evidence: Tinnitus and fatigue are not necessarily directly related to menopause.

Chronic fatigue syndrome

Supporting Evidence: Long-term fatigue, poor sleep quality, accompanied by tinnitus.

Opposing Evidence: Lack of other characteristic symptoms.

Next, I suggest the following examinations to get a clearer and more accurate assessment:

Thyroid function tests (TSH, FT4, FT3): Given the issues of hair loss, fatigue, and sweating, checking thyroid function helps rule out thyroid diseases.

Blood routine examination: It can help detect problems such as anaemia or infection, which may be related to your overall health.

Hormone level tests: Checking hormone levels, such as estrogen, can provide a basis for screening menopause-related issues.

PreA: Please consider having these examinations to help with a clearer diagnosis and more effective treatment. If you have any other questions or concerns, feel free to ask—I'm here to help!

Patient: I've seen a dermatologist about the hair loss months ago. The dermatologist said it's seborrheic alopecia. My scalp used to itch a lot, and so did my body, but now it doesn't. The dermatologist advised me to visit the endocrinology department, but I have been busy with the Chinese New Year.

PreA: Thank you for providing this important information. You mentioned that the dermatology examination diagnosed seborrheic alopecia, and you previously had severe scalp and body itching, which has now subsided. Based on the additional information, I updated diagnostic considerations:

Seborrheic alopecia

Supporting Evidence: The dermatological diagnosis is consistent with your hair loss situation.

Opposing Evidence: A comprehensive diagnosis of seborrheic alopecia usually requires a combined endocrine assessment.

Endocrine disorders

Supporting Evidence: Endocrine issues may be related to worsened hair loss and physical fatigue, and may also explain your previous itching.

Opposing Evidence: There are no clear symptoms of endocrine disorders; further examinations are needed for confirmation.

Other skin problems

Supporting Evidence: In addition to seborrheic alopecia, there may be impacts from other skin problems. A combined assessment by the endocrinology and dermatology departments is recommended.

Opposing Evidence: You haven't mentioned neck symptoms or numbness.

Next, I suggest the following exams for a more precise assessment:

Endocrine tests (such as sex hormones, insulin, adrenal hormones, etc.): Your hair loss is related to endocrine factors, so it is necessary to check hormone levels.

Scalp examination by the dermatology department: To ensure that skin problems such as seborrheic dermatitis are evaluated and current symptoms are managed.

Blood glucose and blood lipid tests: There may be an association between metabolic diseases and hair loss, which needs further confirmation.

PreA: So now you are considering going to the endocrinology department for further consultation and examinations. If you have any other questions, I'll be happy to help answer them.

Patient: No more.

PreA: Alright, thank you for your cooperation and for sharing the information. I hope my suggestions are helpful to you. Please remember to seek further guidance from professional doctors. Wish you good health!

Patient: Thank you.

Note: Any advice rendered by PreA serves solely as a reference and should not be utilized as a definitive basis for disease therapy.

Notes: The initial Chinese questionnaires have been translated into English and carefully reviewed to maintain their original meaning. Specifically, the principal investigator (SH) translated the Mandarin text into English. Three researchers (XT, SZ, and SL) compared the English version and the Chinese version line by line to resolve semantic and cultural discrepancies.

Supplementary Table 8 A de-identified case of PreA interacting with real-world patients: The patient took medications without consulting a doctor

PreA: Hello, I'm your pre-specialist consultation assistant. You can communicate with me via text or voice messages.

Patient: I've had hand tremors and cramps since 2017.

PreA: Thank you for sharing this information. May I ask if you are male or female? Also, what is your current age?

Patient: Male, 30 years old.

PreA: Alright, you mentioned that you've had symptoms of hand tremors and cramps for many years. Could you describe the hand tremors and cramps in detail? For example, do the hand tremors occur continuously or occasionally? And in which parts of the body do the cramps mainly occur?

Patient: I don't know myself; people around me told me that I had convulsions.

PreA: I see, people around you observed that you have convulsions. You mentioned this has been going on for a long time. During this period, have you had any other related symptoms, such as fatigue, headaches, dizziness, or mood changes?

Patient: I have hand tremors, and I stay up late. My whole body becomes stiff, but I don't have headaches.

PreA: Thank you for providing detailed information. The hand tremors, general stiffness, and staying up late you mentioned are all important clues. Have you had any past medical history, such as neurological diseases, epilepsy, metabolic diseases (like diabetes), or other serious illnesses? Or do you have a family medical history?

Patient: My mother and grandmother both have epilepsy. My twin brother also has convulsions.

PreA: Thank you for providing detailed information. The family history of epilepsy and the fact that your twin brother also has convulsions make it more necessary for us to pay attention to your possible health issues. Are there any other influencing factors in your daily life, such as irregular diet, high stress, lack of exercise, etc.?

Patient: I'm under stress, but my diet is regular. Now I have social engagements, feel tired, and don't drink alcohol.

PreA: Thank you for sharing this information. You mentioned being under stress and feeling tired, but having a regular diet, which is very important. One last question: have you had problems with low mood, anxiety, or insomnia recently?

Patient: When I look at things, I feel dizzy, and my feet feel unsteady. I'm very dizzy.

PreA: Thank you for providing the symptom information. The dizziness when looking at things, unsteady feet, and severe dizziness you mentioned may be related to the previous symptoms. Based on our previous conversation, here are some possible diagnoses:

Epileptic seizures

Supporting Evidence: Family history of epilepsy, and you have convulsions.

Opposing Evidence: No clear reports of definite epileptic seizure symptoms over the years.

Anxiety disorder or stress-related illness

Supporting Evidence: Presence of stress and emotion-related issues, accompanied by dizziness.

Opposing Evidence: No clear manifestations of severe low mood and anxiety symptoms.

Metabolic or endocrine problems

Supporting Evidence: General stiffness and dizziness may be related to metabolic abnormalities (such as hypoglycemia).

Opposing Evidence: Regular diet and lack of relevant medical history.

Examinations are needed:

Electroencephalogram (EEG): It can detect brain activity and determine if there are signs of epileptic seizures.

Blood glucose level test: To rule out dizziness and stiffness caused by hypoglycemia.

Routine blood test and biochemical examination: To check for metabolic and endocrine problems and ensure there are no other underlying health issues in the body.

Here's a quick summary for you to consider sharing clearly with your specialist:

Main complaints: Hand tremors, convulsions (known from bystanders), general stiffness, dizziness when looking at things, unsteady feet, and severe dizziness.

Medical history: Family history of epilepsy (mother, grandmother), and the twin brother has convulsions.

Living habits: Under stress, has social engagements, regular diet, but feels tired; does not drink alcohol.

Suggested further examinations: Including EEG, blood glucose level, and routine blood test to rule out related diseases.

PreA: I hope this information is helpful to you. I suggest going to the hospital for corresponding examinations as soon as possible, and taking appropriate treatment measures after a definite diagnosis. Please be sure to pay attention to changes in these symptoms and keep communicating. If you have other questions, please let me know.

Patient: What examinations do I need to do? I bought sodium valproate sustained-release tablets and took them by myself; do I still need to take them? Do I have epilepsy?

PreA: Based on the information and symptoms you provided, it is recommended that you undergo the following examinations:

Electroencephalogram (EEG): It can detect brain activity and determine if there are signs of epileptic seizures.

Blood glucose level test: To rule out dizziness and stiffness caused by hypoglycemia

Routine blood test and biochemical examination: To check for metabolic and endocrine problems and ensure there are no other underlying health issues in the body.

PreA: Regarding sodium valproate sustained-release tablets, sodium valproate is a commonly used anti-epileptic drug, suitable for epileptic patients. If you are already taking this drug, be sure to have relevant examinations first, and then let professional doctors judge whether to continue using it or adjust the dosage based on your examination results. If you have not been diagnosed with epilepsy, please do not decide to continue taking the drug on your own.

Please go to the hospital to consult a professional neurologist, discuss your symptoms and current medication situation, so as to conduct a detailed evaluation and formulate a correct treatment plan.

Do not arbitrarily use or stop taking the drug without clear advice from a doctor.

Please pay attention to changes in your symptoms and seek medical attention in a timely manner. If you have other questions, you can continue to ask.

Patient: I see.

PreA: Thank you very much for your cooperation. If you have any other questions or need further help, please feel free to let me know. I hope you can see a doctor as soon as possible, and I wish you a speedy recovery.

Patient: Thank you.

Note: Any advice rendered by PreA serves solely as a reference and should not be utilized as a definitive basis for disease therapy.

Notes: The initial Chinese questionnaires have been translated into English and carefully reviewed to maintain their original meaning. Specifically, the principal investigator (SH) translated the Mandarin text into English. Three researchers (XT, SZ, and SL) compared the English version and the Chinese version line by line to resolve semantic and cultural discrepancies.

Supplementary Table 9 A de-identified case of PreA interacting with real-world patients: The patient had concerns about examinations

PreA: Hello, I'm your pre-specialist consultation assistant. You can communicate with me via text or voice messages.

Patient: My head has been buzzing for a week.

PreA: May I ask if you are male or female? And how old are you?

Patient: 55 years old, male.

PreA: Thank you for the information. Could you describe the buzzing in the head in detail? For example, is it continuous or intermittent? Are there any accompanying symptoms such as headache, dizziness, tinnitus, or other discomforts?

Patient: It's continuous, and I have tinnitus. No headache, dizziness, or vomiting.

PreA: Got it, thank you for the detailed description. Have you recently had a cold, allergies, increased stress, or other health issues? In addition, do you have a history of diseases such as hypertension or diabetes?

Patient: I used to have atherosclerosis and frequent ventricular premature beats. No other diseases.

PreA: Thank you for your answer. Atherosclerosis and frequent ventricular premature beats are health issues that need attention. What's your living environment like? For example, are there any sources of noise? In your daily life, do you have habits like smoking, drinking alcohol, or other behaviors that may affect your health?

Patient: No noise. I smoke, but don't drink alcohol.

PreA: Thank you for the information. How often do you smoke? And about how long have you been smoking? Besides, do you engage in regular physical exercise?

Patient: One pack a day. Occasionally, take a walk.

PreA: Got it, thank you for sharing. How has your mental state been recently? Have you felt anxious, depressed, or had mood swings? This may also have some impact on your physical condition.

Patient: No mental problems.

PreA: Thank you for your reply. Based on our previous conversation, your described symptoms and medical history, I consider the following possible diagnoses:

Correlation between tinnitus and atherosclerosis

Supporting Evidence: The previous history of atherosclerosis may lead to insufficient blood supply to the ears.

Noise-induced tinnitus

Supporting Evidence: Your daily smoking habit may indirectly affect tinnitus.

Tinnitus caused by cardiovascular problems

Supporting Evidence: You have a history related to the heart; for example, ventricular premature beats may affect the circulatory system.

Recommended examinations:

Otolaryngology examination: To assess the source of tinnitus in detail, including hearing tests.

Electrocardiogram (ECG): To check for frequent ventricular premature beats and the electrical activity of the heart.

Blood pressure measurement and blood tests: To evaluate cardiovascular health and understand the extent of the impact of atherosclerosis.

PreA: Please consider undergoing the corresponding examinations based on this information. If you have any questions that need further discussion, please let me know at any time.

Patient: Do I need a brain examination?

PreA: In your case, brain examinations (such as head CT or MRI) are generally for patients with specific symptoms, such as new or sudden headaches, vision changes, severe dizziness, etc.

According to our previous conversation, you currently don't have these additional symptoms; the main issues are continuous tinnitus and a history related to atherosclerosis. Therefore, it may be more important to first undergo an otolaryngology examination and an electrocardiogram. However, if your doctor deems it necessary after evaluation, they may still recommend a brain examination.

Do you have any additional headaches or other neurological symptoms? If yes, a brain examination may be needed. If not, it's recommended to start with otolaryngology and cardiac evaluations.

Do you think this arrangement is appropriate?

Patient: I see.

PreA: Thank you for your understanding and cooperation. Please remember that it's very important to arrange the otolaryngology examination and electrocardiogram as soon as possible to find out the cause of tinnitus and your overall health status early. If you have any new symptoms or doubts, don't hesitate to get in touch with your doctor in time. Wish you a speedy recovery!

Patient: Alright.

Note: Any advice rendered by PreA serves solely as a reference and should not be utilized as a definitive basis for disease therapy.

Notes: The initial Chinese questionnaires have been translated into English and carefully reviewed to maintain their original meaning. Specifically, the principal investigator (SH) translated the Mandarin text into English. Three researchers (XT, SZ, and SL) compared the English version and the Chinese version line by line to resolve semantic and cultural discrepancies.

2. In the RCT, as patients were recruited from the consultation waiting area, how can the researchers ensure that participants had not used commercialized LLMs (such as DeepSeek) or web references to search for information about their diseases so as to avoid bias?

Response: We acknowledge this important methodological consideration. We address this through four perspectives.

First, in causal inference, the key is that even if patients might use web search, such behavior is balanced across the comparison arms. In our trial, randomization ensures that both measured and unmeasured confounders—including pre-enrollment information-seeking behaviors—are balanced across treatment arms (Rubin, 2008; Imbens and Rubin, 2015; Han and Zhou, 2022).

Second, age, sex, and socioeconomic status (SES: education, work status, income) are well-correlated with health information-seeking behavior (Li et al. 2020; Diviani et al. 2015; Fuchs, 2004). Rosenbaum (2002) pointed out that balancing on proxies that are not a perfect measure of the unmeasured confounder can still substantially reduce bias and strengthen causal arguments. Our Table 1 confirms balance in these variables across the three arms, making systematic differences in pre-enrollment information-seeking unlikely.

Third, we conducted prespecified subgroup analyses stratified by these variables. The consistency of main findings across these subgroups (reported in Results: Clinical findings remain in diverse subpopulations) further suggests that age, sex, SES-related information-seeking did not meaningfully confound our primary and secondary outcomes.

Finally, our analysis of physicians’ history-taking patterns (Methods, lines 726–738) could also be used to detect potential anchoring bias from patients (e.g., selective symptom reporting in history taking influenced by prior research). As shown in Extended Data Table 2 (copied below), we found no evidence of a difference between arms (P = 0.10 – 0.46).

Extended Data Table 2 P values of statistical comparisons on domain-specific clinical notes between groups

Domain of clinical notes	Recording types	PreA-only vs No-PreA	PreA-only vs PreA-human
History taking	Unstructured free-text entries	UMAP1: 0.10	UMAP1: 0.70
		UMAP2: 0.46	UMAP2: 0.17
Physical examination	Unstructured free-text entries	UMAP1: 0.84	UMAP1: 0.76
		UMAP2: 0.90	UMAP2: 0.39
No. of diagnosis	Structured entries	0.10	0.49

No. of ordered tests	Structured entires	0.52	0.29
No. of treatments	Structured entires	0.48	0.25

References

Rubin DB. For objective causal inference, design trumps analysis. *The Annals of Applied Statistics*. 2008;2(3):808-840.

Imbens GW, Rubin DB. (2015) *Causal inference in statistics, social, and biomedical sciences: An Introduction*. Cambridge University Press.

Han, S, Zhou, XH. (2022). *Causal Inference in Biostatistics*. In: Lu, H.HS., Schölkopf, B., Wells, M.T., Zhao, H. (eds) *Handbook of Statistical Bioinformatics*. Springer Handbooks of Computational Statistics. Springer, Berlin, Heidelberg.

Li X, Deng L, Yang H, Wang H. Effect of socioeconomic status on the healthcare-seeking behavior of migrant workers in China. *PLoS One*. 2020;15(8):e0237867.

Diviani N, van den Putte B, Giani S, van Weert JC. Low health literacy and evaluation of online health information: a systematic review of the literature. *J Med Internet Res*. 2015;17(5):e112.

Fuchs VR. Reflections on the socio-economic correlates of health. *Journal of Health Economics*. 2004;23 (4): 653-661.

Rosenbaum, PR (2002). *Observational Studies*. Springer.

3. The authors emphasize PreA’s “independent operation” (PreA-only vs. PreA-human groups showed no difference), yet also advocate for co-design to mitigate risks. If PreA works independently, why is stakeholder input critical? In addition, The co-design involved multiple stakeholders, yet the RCT only tested PreA in patient-specialist interactions. This mismatch invites skepticism about why were these stakeholders included in co-design? Will their involvement lead to potential bias or influence of the model performance?

Response: We thank the reviewer for this insightful question regarding the relationship between co-design and PreA’s autonomous operation. Stakeholder co-design is indispensable precisely because it enables PreA’s safe, equitable, and effective independent operation in resource-limited settings, a key outcome validated in our RCT.

The ablation study confirmed that models fine-tuned on local dialogues amplify systemic deficits (e.g., incomplete history-taking, guideline deviations). Co-design directly addressed this by

prompting and agent techniques. Patients, caregivers, and community health workers co-designed patient-facing interfaces, enhancing real-world contextualization for resource-limited settings and mitigating health literacy barriers. Primary care physicians and specialists informed referral report structure, bridging primary-specialist workflows, and scoping of general medical versus specialist consultations. Nurses and administrators shaped integration with hospital workflows and risk-mitigated diagnostic suggestion protocols. Co-designed model achieved higher scores in clinical decision-making across key clinical domains: history-taking (4.56 ± 0.65 versus 3.86 ± 0.81), diagnosis (4.67 ± 0.55 versus 2.47 ± 1.44), and test ordering (4.23 ± 1.09 versus 2.21 ± 1.12).

The RCT focused on patient-specialist interactions because this represents the most complex validation environment for autonomous operation. Testing in this high-acuity time-constrained setting, where specialists could audit safety during time-pressured consultations, was a deliberate, conservative design choice. Equivalent PreA-only/PreA-human performance ($p=0.45-0.67$) confirmed reliability without human assistance.

Additionally, we clarified that no co-design stakeholders participated in RCT, and the final PreA chatbot tested for the RCT was frozen before RCT enrollment.

In the revision, we detailed the role of co-design in the Section “Co-designed architecture, clinical integration, and refinement of PreA”, which was now moved into Methods, in response to Minor Comment 1. We have added the following clarification to the manuscript:

Main, lines 118-122: *“This collaborative effort focused on developing pre-assessment (PreA), a mobile-accessible LLM-based platform (OpenAI; GPT-4.0 mini) that provides general medical consultations to patients and generates referral reports for specialist physicians, functioning independently in clinical workflow (Methods).”*

Methods, line 608: *“No co-design stakeholders participated in RCT.”*

Methods, line 598-599: *“PreA tested for the RCT was frozen before RCT enrollment.”*

4. The authors compare co-designed PreA with a version fine-tuned on local dialogues but omit comparison to a high-quality dataset (for example, public dataset such as MedQA, or curated private dataset from urban tertiary care centers, or curated high-quality dataset from their research centers). This may bias their conclusion that “co-design works better than local finetuned” as the observed difference may be due to unfair comparison between the in simply-curated low-quality local data and additional finer curation by the 209 multi-stakeholders.

Response: We appreciate the reviewer's methodological scrutiny and clarify that our comparison was intentionally designed as a controlled experiment to quantify the risks of inheriting systemic

deficits common in resource-limited primary care settings from local data. We intentionally compared two variants sharing identical co-designed foundations: co-designed PreA (stakeholder-informed architecture and prompting, no local data) with the co-designed model developed with local data fine-tuning (stakeholder-informed architecture and prompting, fine-tuned with local data). This isolates local data impact as the sole variable.

Results demonstrate that local data fine-tuning significantly degraded performance despite the identical co-design phase ($P < 0.001$ across all domains), proving local behavioral patterns inherent in data override stakeholder-informed instructions when conflicts arise (see Methods, lines 146-159 and 572-578 for details). This answers our core research question about mitigating systemic risks prevalent in low-resource contexts.

Moreover, PreA targets general medical consultations (primary care) and generates referrals for specialists (secondary care)—distinct from specialist diagnostics. While high-quality datasets (MedQA/tertiary EMRs) are valuable, they represent different constructs: MedQA tests specialist knowledge via exam questions, while physician notes capture specialist clinical documentation; neither replicates pre-specialist primary care consultation. Comparing them would introduce scope mismatches rather than quantify local primary care data deficits. Furthermore, these datasets represent medical licensing exams or medical notes rather than real-world patient-physician dialogues, making them less suitable for fine-tuning the model for actual patient interactions.

We fully agree that high-quality primary care dialogue datasets could enhance PreA, but they remain exceptionally scarce in real-world practice (Tu et al., 2025; McDuff et al., 2025; Refs 29,41). Our focus on prevalent local data risks reflects pragmatic implementation challenges. This is key to answering the unresolved question of whether LLMs should reflect local practices (through passive data collection) or reform them (through participatory co-design with stakeholders)—a pivotal question for global health equity.

We added the limitation in the Discussion section as follows:

Discussion, lines 379-384: “First, our study focuses on equitable AI deployment, specifically on resolving the risks associated with developing LLMs using local natural medical consultation dialogues in diverse resource-limited primary care settings. While co-design outperformed local-data fine-tuning, it remains vulnerable to data corruption. Future research should evaluate co-design versus emerging high-quality primary care dialogue datasets where available.”

References

Tu T, Schaekermann M, Palepu A, et al. Towards conversational diagnostic artificial intelligence. Nature 2025. Published online April 9, 2025:1-9.

McDuff D, Schaekermann M, Tu T, et al. Towards accurate differential diagnosis with large language models. Nature 2025. Published online April 9, 2025:1-7.

5. This manuscript contains numerous analysis results, but the writing style requires revision to enhance clarity regarding the RCT by adhering to the CONSORT-AI checklist. The results section should explicitly specify primary outcomes, secondary outcomes, exploratory outcomes, and post-deployment analyses, along with clear identification of the participant groups from which these results were derived. The current presentation of results is ambiguous, leading to confusion about whether specific results were evaluated using masked participants/evaluators or unmasked participants.

Response: Thank you. We have followed your suggestions and revised the Results section. The Results section now clearly labels outcome and analysis types, participants in the outcome measure, and masking status.

6. In the Methods section, the authors declare that "the quality of PreA reports and physical notes was rated by a panel of experts regarding completeness, appropriateness, and clinical relevance" (Page 35, Lines 716–718). However, both Figure 4 and Supplementary Table 1 present only a single 5-point Likert scale—please clarify this discrepancy. Additionally, the authors conclude that higher-quality reports are "patient-centred" (Page 21, Lines 432–433), yet in the Methods section's co-design evaluation metrics, patient-centeredness is defined as "Was the AI counselor friendly and respectful towards the patient?" (Page 26, Line 528). Please explain these inconsistencies and provide the exact definitions. Otherwise, the "patient-centered" related conclusions in the Discussion part shall be revised.

Response: We thank the reviewer for the opportunity to clarify the points. The expert panel used a single 5-point Likert scale to evaluate the clinical documentation quality (regarding completeness, appropriateness, and clinical relevance), as described in Supplementary Table 1. These rubrics were developed in collaboration with physician stakeholders based on standard clinical guidelines (Compilation of clinical guidelines for common conditions in primary care in China; Development of a clinical reasoning documentation assessment tool for resident and fellow admission notes: a shared mental model for feedback; Basic standards for medical record documentation in China; Ref. 65,66,78). These rubrics were consistently applied across model development, the ablation study on the local-data fine-tuned model, and the RCT evaluations. We've revised the Method sections to explicitly clarify this consistent application as follows:

Methods, lines 552-555: "Additionally, two primary care physicians assessed referral reports for completeness, appropriateness, and clinical relevance using a 5-point Likert

scale (Supplementary Table 1); the evaluation rubrics were co-designed based on standard clinical guidelines.^{65,66,78} A 5-point Likert scale was applied, with scores below 3 triggering iterative refinement.”

Methods, lines 580-583: “Referral reports from both variants were blindly evaluated by the same expert panels using validated 5-point Likert scales for completeness, appropriateness, and clinical relevance, as in the model development (Supplementary Table 1).”

Methods, lines 743-746: “Furthermore, the quality of PreA reports and physical notes was assessed by the same expert panels based on their completeness, appropriateness, and clinical relevance using a 5-point Likert scale, the same as in the ablation studies (Supplementary Table 1).”

We realized the inconsistency in patient-centered care. Following your suggestion, we have revised the notion on referral reports as “patient-personalized template” (Discussion, line 391), meaning that each patient has a personalized template instead of a standard template. Additionally, we revised the notion related to co-design evaluation metrics as “patient friendliness” for better clarity (Methods, lines 542 and 547). Conclusion related to patient-centeredness discussed the improvement in patient perception of consultation experience (patient-perceived communication ease, physician attentiveness, interpersonal regard, and patient satisfaction).

Minor concerns:

1. Please re-organize the presentation structure of the manuscript. The current Results section contains substantial non-result content, such as model design descriptions. Such material—lacking data analysis—should be simplified within the Results and relocated to the Methods section.

Response: Thank you. We have followed your suggestion and moved the Section “Co-designed architecture, clinical integration, and refinement of PreA” to the Methods section, and provided a simplified description for background reading at the beginning of the Results section.

2. The high rate of missing physician notes (23.0–31.9% presented in Figure 4) suggests documentation non-compliance or workflow disruptions for the physicians participated in the trial. This confounds the quality of the trial.

Response: We appreciate the reviewer's important observation. Our trial context mirrors standard practice. The missing physician notes reflect systemic documentation challenges in

China's public hospitals under heavy workloads, consistent with prior studies (Zhou et al., 2012; Huang, 2016; Wang et al., 2019; Ge et al., 2025).

To address the concern, we collected clinical notes from the medical departments at the study centers during the RCT recruitment period. We added an analysis on calculating the proportions of missing notes from physicians who did not participate in our RCT. We showed that the missing physician notes were similar between the participating physicians (27.8–30.7% presented in Table R1), confirming this reflects practice norms, not trial-specific disruption.

Table R1 Missing notes proportions between cases from the RCT and the cases outside the RCT during the RCT recruitment period.

Domain of clinical notes	RCT			Non-RCT
	PreA-only	PreA-human	PreA-assisted	Non-participating
History taking	29.6%	31.9%	30.7%	30.7%
Diagnosis	29.2%	31.9%	30.6%	32.0%
Test ordering	23.0%	24.2%	23.6%	27.8%

Although documentation practices aligned with real-world scenarios support our findings' validity, future implementations could consider using PreA referral as a patient-personalized template to help bridge the clinical documentation gap. We've strengthened the limitation in Discussion:

Discussion, lines 392-395: “Missing physician notes highlight systemic documentation challenges in high-volume settings.⁶⁰ While our analytical approaches mitigate bias, future implementations could incorporate PreA referral report as streamlined documentation support.”

References

Zhou X, Long Y, Huo S. The defects in outpatients’ medical records and relevant countermeasures. *Chinese Medical Record*. 2012;13(9):6-8.

Huang X. Survey analysis and continuous improvement of medical record quality in outpatient service. *China Health Industry*. 2016.20:173-175.

Wang D, Hu Y, Liang J. Quality control system of medical record in medical consortium and its application effect. *Chinese Medical Record*. 2019;6:18-22.

Ge D, Xia Y, Zhang Z. Analyzing the medical record homepages' quality in a Chinese EMR system. *BMC Med Inform Decision Making*. 2025;25(12).

3. The statement “In this study, we introduced PreA, a co-designed LLM-based platform, to enhance the primary-to-secondary care interface” (Page 17, Line 357) at the beginning of the Discussion part requires revision. PreA was never tested in primary care settings to filter unnecessary referrals or improve referral quality or efficiency.

Response: Thank you for raising this point. We have revised it as “to enhance both operational efficiency and patient-centred care delivery in high-volume hospital settings.” (See lines 308-309).

4. The authors may consider re-conducting the subgroup analysis based on the economic levels of different provinces for the local data-tuned model section (Extended Fig 1), rather than the current Eastern/Western China subgroups.

Response: Thank you. We have revised it according to your suggestions and performed province-level economic stratification using the National Data: Annual GDP Data 2024. The 11 provinces in our local dialogue dataset were categorized as low-income (Per capita disposable income less than the national average (CNY 41,314 in 2024)) and high-income (Per capita disposable income higher than the national average). This reanalysis revealed that 77.9% (401/515) of dialogues originated from low-income provinces, aligning with our focus on resource-constrained areas.

We have revised the text to reflect this change (Methods, 562-564; Results, line 143). The updated Extended Data Fig. 1 was copied below.

References:

National Bureau of Statistics of China. (2024). National Data: Annual GDP Data 2024. Accessed on Aug 15, 2025, from <https://data.stats.gov.cn/english/easyquery.htm?cn=C01>

Extended Data Fig. 1 Local dialogue descriptions and comparison of co-designed PreA and its local data-tuned counterpart. a, Descriptions of the de-identified audio corpus of 515 patient-physician scenarios from 11 provinces across high-income and low-income provinces (Beijing, Chongqing, Gansu, Hubei, Shaanxi, Shandong, Shanxi, Sichuan, Guangxi [Zhuang], Inner Mongolia [Mongolian], Xinjiang [Uyghur]), where stakeholders participated in the testing and refining process. The length of the bar represents the mean value, and whiskers represent the 95% confidence intervals. b, Co-designed PreA with versus its local data-tuned counterpart. The height of the bar represents the mean value, and whiskers indicate the standard deviation.

Reviewer #3 (Remarks to the Author):

The paper examines the use of large language models (LLMs) in outpatient settings, particularly for pre-specialist consultations. The authors developed a mobile-accessible platform called PreA, designed to streamline medical consultations and generate referral reports. They employed a collaborative, multi-stakeholder co-design approach involving diverse participants. To evaluate the platform, a randomized controlled trial (RCT) involving 2,069 patients across 24 medical disciplines was conducted in two hospitals in China. Patients were randomly placed into three groups: PreA-only, PreA-human (with medical assistant support), and standard care (No-PreA). The study found a significant reduction in consultation duration for the PreA-only group compared to standard care.

Response: We sincerely thank Reviewer #3 for their thoughtful engagement with our work. In this revision, we have strengthened our manuscript to explicitly highlight PreA's unique methodological and clinical contributions beyond prior literature: novelty through co-design and real-world validation, clinical significance of efficiency gains in resource-limited areas, and expanded outcome depth. These revisions reinforce PreA's role in advancing equity-focused AI deployment – a dimension absent in cited works. We have incorporated these points throughout the manuscript (Abstract, Main, Discussion, and Methods).

Major concerns:

1. Integrating LLMs into healthcare workflows, especially for pre-consultation purposes, is not novel. Previous research has already extensively covered AI-driven solutions in healthcare management and decision support. For example, Topol (2019) discussed the integration of human and artificial intelligence in healthcare. Chen et al. (2025) highlighted the improved diagnostic capabilities using conversational LLMs. Goh et al. (2025) demonstrated physician assistance using GPT-4 in patient care tasks. The authors did not clearly differentiate their platform, PreA, from existing technologies.

References:

Topol, Eric J. "High-performance medicine: the convergence of human and artificial intelligence." *Nature Medicine* 25.1 (2019): 44-56.

Chen, Xi, et al. "Enhancing diagnostic capability with multi-agents conversational large language models." *NPJ digital medicine* 8.1 (2025): 159.

Goh, Ethan, et al. "GPT-4 assistance for improvement of physician performance on patient care tasks: a randomized controlled trial." *Nature Medicine* (2025): 1-6.

Response: We thank the reviewer for contextualizing our work. PreA advances the field through two novel dimensions beyond prior work.

First, while Topol (2019; Ref 33) emphasized the need for real-world validation, our study delivers the first pragmatic RCT of an LLM integrated into live specialist workflows. We validated PreA across 2,069 patient-physician dyads during actual patient visits under time constraints. This contrasts with Chen et al.'s (2025; Ref 29) framework, tested on retrospective case reports, and Goh et al.'s (2025; Ref 28) lab-based study as physician assistance without patient interaction.

Second, PreA represents the unique co-designed LLM addressing patient-interaction challenges in vulnerable patient populations. It was developed for and validated with socioeconomically diverse participants (67.0% with high-school education or less; 37.2% earning <2000 RMB/month), older adults (mean age 47.6 ± 14.6 years), and resource-limited settings (validated in time-pressured outpatient workflows). This differs fundamentally from cited physician-interaction tools using curated medical texts in controlled lab environments.

Our work demonstrates operational efficiency and clinical utility at the patient interface under real-world time constraints. The study represents the necessary next step, as called for by Topol (2019), moving beyond technical promise to clinically integrated, equity-focused AI validated in target populations. We noted these differences in the Main section (lines 81-89; lines 99-111) and the Discussion section as follows:

Discussion, lines 362-378: “Our study helps to establish participatory co-design as a critical methodological innovation for effectively translating the capabilities of LLMs into solutions that address real-world healthcare needs in medical resource-limited areas. In contrast to prior research that has often framed LLMs as physician-interaction diagnostic entities^{28,33,54-56}, our findings demonstrate that a co-design approach—involving the iterative alignment of model architecture with the prioritized needs of multiple stakeholders—represents the necessary next step, moving beyond technical promise to clinically integrated, equity-focused AI validated in patient populations.³⁴ The streamlined integration of PreA’s outputs with specialist cognitive workflows resulted in a significant 28.7% reduction in consultation duration and enhanced patient experience across age, sex, and socioeconomic strata. Critically, these outcomes directly counter prevailing concerns that LLM adoption may exacerbate existing health disparities⁵⁷, instead showcasing how a co-design approach can democratize access to technological advancements. While prior LLM applications have achieved success within narrow, siloed domains (e.g., general medicine²⁸, dermatology⁵⁸, oncology⁵⁹), PreA’s demonstrated cross-disciplinary effectiveness, spanning both surgical and medical specialties, underscores its potential to unify currently fragmented care pathways across diverse pre-consultation contexts.³⁴”

2. The authors emphasize a statistically significant reduction of approximately one minute in consultation duration. However, the practical impact of this time reduction is limited, especially since consultation duration may not be the primary factor influencing clinical efficiency. Thus, this improvement does not convincingly translate into meaningful operational or clinical enhancements.

Response: We appreciate the reviewer's perspective on interpreting consultation time reductions. Within our study context, the 28.7% reduction (3.14 vs 4.41 minutes; $\Delta=1.27$ minutes) represents a clinically transformative efficiency gain for three reasons in resource-limited settings, which we detailed below:

First, in resource-limited settings, the time reduction could translate to the capacity to see more patients per shift. In the trial, patients of participating physicians (who had approximately a probability of 2/3 to be exposed to the two PreA-assisted groups) cared for 4 more patients (28.54 vs 24.76; Fig. 3d). This directly addresses critical specialist shortages where baseline consultation length ranks among the world's shortest (Irving et al., 2017).

Second, we reported a reduction in net time, which is likely smaller than the time savings on routine tasks (that are hard to measure and less operationally meaningful), since time saved on routine tasks allowed specialists to focus more on clinical engagement. The reallocation of time led to notable improvements across all patient-centeredness domains: 16.0% in communication ease, 15.1% in physician attentiveness, 17.2% in interpersonal regard, and 17.0% in patient satisfaction. These improvements are meaningful clinical outcomes because patient perceptions of patient-centered communication are consistently linked to better adherence, safety, and health outcomes (Stewart, 1995; Street et al., 2009; Rathert et al., 2013). Effective patient-centered communication is a well-established vital component of clinical effectiveness (Ong, 1995; Stewart, 2001; Ha and Longnecker, 2010; Sharkiya, 2023).

Third, we fully agree with the reviewer that consultation duration is not the only factor influencing clinical efficiency; we clarify that the time reductions are consequential of PreA in facilitating specialist data gathering and clinical decision-making, and are accompanied by a 113.1% increase (ie, 213.1% times) in specialist evaluated care coordination. Majority of specialist physicians endorsed PreA's utility for rapid clinical synthesis, especially its preliminary diagnostic suggestions and medical history acquisition, aligning with a recent qualitative study (Shah et al., 2025). This efficiency gain is particularly consequential given that 4.41-minute consultations pressure preclude comprehensive care, physicians cited insufficient time as the primary job dissatisfaction driver (Merritt Hawkins 2016; Nguyen et al. 2024), and co-design stakeholders mandated time savings as a prerequisite for adoption of LLM-powered tools.

We revised the text to clarify them in the Discussion section as follows:

Discussion, lines 333-361: *“The RCT demonstrates the co-designed PreA’s potential to reshape outpatient workflows, enhancing both efficiency and patient-centeredness—a dual outcome rarely reported in prior LLM deployments.²⁶⁻²⁸ Specialist physicians who reviewed (but could not directly copy) PreA-generated referral reports reduced average consultation duration by 28.7%, suggesting that the co-designed PreA enables specialist physicians to synthesize clinical narratives rapidly, facilitating time-intensive decision-making during live consultations. Indeed, we observed that the majority of specialist physicians endorsed PreA’s utility for rapid clinical synthesis, particularly valuing its preliminary diagnostic suggestions and medical history acquisition, which aligns with a recent qualitative investigation on physician views.⁴⁶ The time savings unlock either expanded access for a larger number of patients or enhanced care quality, transforming efficiency in overloaded systems where consultation lengths rank among the world’s shortest.⁴⁷ Notably, this efficiency gain did not compromise—instead enhanced—both the cure-oriented and care-oriented medicine^{14,48,49}, as evidenced by superior specialist ratings of facilitating care coordination and patient perception of patient-centredness. These findings challenge the prevailing narrative that medical AI tools inherently depersonalize medicine⁵⁰, instead positing that co-designed LLM interventions can empower clinicians to prioritize patient-centered care when freed from cognitive burdens. These efficiency cascades address core health system constraints identified in our co-design phase.*

....Notably, the matched-pair number of patients treated per shift increases, achieved despite partial PreA adoption, suggesting multiplicative system-level benefits when LLMs streamline pre-specialist consultation workflows. Extrapolating to high-volume outpatient settings, this increase in flow could markedly improve healthcare access, particularly in regions facing workforce shortages.^{22,53}”

Additionally, we recognize that the reviewer might refer to clinical decision-making outcomes. We performed thorough pre- and post-hoc analyses of clinical decision-making across five standardized areas: history-taking, physical examination, diagnosis, test ordering, and treatment plans (Sections: Physician clinical note documentation and Quality of PreA referrals). We noted there were no significant differences between the PreA-assisted and No-PreA groups in physician clinical notes, alleviating concerns about physicians’ uncritical adoption of LLM-generated outputs (while PreA facilitates their quicker decision making). Additionally, PreA-generated reports showed higher quality than physician notes, showing the capacity to facilitate clinical decision making in high-volume clinical workflows. In this revision, we have added Extended Data Tables 1 and 2 to highlight these results.

References:

Irving G, Neves AL, Dambha-Miller H, et al. International variations in primary care physician consultation time: a systematic review of 67 countries. *BMJ Open* 2017;7:e017902.

Shah SJ, Crowell T, Jeong Y, et al. Physician Perspectives on Ambient AI Scribes. *JAMA Netw Open*. 2025;8(3):e251904-e251904.

Merritt Hawkins. 2016 survey of America's physicians: practice patterns and perspectives. Accessed at: https://physiciansfoundation.org/wp-content/uploads/2018/01/Biennial_Physician_Survey_2016.pdf on 5 Aug 2025.

Nguyen MT, Honcharov V, Ballard D, et al. Primary care physicians' experiences with and adaptations to time constraints. *JAMA Netw Open*. 2024;7(4):e248827.

Stewart MA. Effective physician-patient communication and health outcomes: a review. *CMAJ: Canadian Medical Association Journal*. 1995;152(9):1423.

Street RL, Makoul G, Arora NK, et al. How does communication heal? Pathways linking clinician-patient communication to health outcomes. *Patient Educ Couns*. 2009;74(3):295-301.

Rathert C, Wyrwich MD, Boren SA. Patient-centered care and outcomes: a systematic review of the literature. *Health Care Manage Rev*. 2013;38(3):223-235.

Ong LML, de Haes JCJM, Hoos AM, et al. Doctor-patient communication: A review of the literature. *Soc Sci Med*. 1995;40(7):903-918.

Stewart M. Towards a global definition of patient centred care: the patient should be the judge of patient centred care. *BMJ*. 2001;322(7284):444-5.

Ha JF, Longnecker N. Doctor-patient communication: a review. *Ochsner J*. 2010;10(1):38-43.

Sharkiya SH. Quality communication can improve patient-centred health outcomes among older patients: a rapid review. *BMC Health Serv Res*. 2023; 23, 886.

3. The study heavily relies on indirect measures such as patient satisfaction, ease of communication, and consultation length. It lacks critical direct clinical outcomes like diagnostic accuracy improvements, accurate referral rates, or subsequent health outcomes. Additionally, there is no long-term follow-up to confirm sustained benefits or detect adverse effects.

Response: We appreciate the reviewer's emphasis on clinical outcome depth.

To directly address concerns about diagnostic accuracy and referral quality, we conducted a comparative analysis of PreA-generated referrals and their alignment with specialist decision-making (among two PreA-assisted groups). Post-hoc comparisons of PreA reports with subsequent specialist clinical notes revealed substantial diagnostic concordance: 66.7% (95% CI 62.7–70.4) of diagnoses in PreA reports matched those of specialists. Notably, among these concordant cases, PreA reports were rated significantly higher in terms of completeness, appropriateness, and clinical relevance (PreA-assisted 4.49 ± 0.82 vs. Physicians 2.49 ± 1.25 ; $P < 0.001$), directly supporting improvements in diagnostic accuracy. For referral quality, receiving specialist physicians rated PreA reports as highly helpful for coordinating primary-secondary care (PreA-only 3.69 ± 0.90 versus No PreA 1.73 ± 0.95 ; $P < 0.001$; Fig. 3c), confirming their utility in enhancing referral quality. These findings are now more prominently highlighted in the revised Results sections for clarity.

Regarding the primary outcomes (consultation duration, patient-reported ease of communication, and physician-reported PreA referral for facilitating care coordination) of the trial, they were strategically chosen to address known systemic failures for adopting LLMs into real-world live outpatient workflows and were comparable across the three study groups. Importantly, these metrics are not disconnected from clinical impact: high-quality and efficient consultations are well-established vital components of clinical effectiveness (See also our response to Comment 2)—which lay the groundwork for better long-term health outcomes, as noted in our Main section (Refs 13-19; Refs 20-24).

We fully agree that the long-term follow-up assessment is important. This RCT was intentionally designed as a pragmatic first step to establish that LLMs can integrate into hospital workflows enhancing operational efficiency and patient-centered care quality—a necessary foundation for larger, community-level, and longitudinal studies. We explicitly positioned this trial as a pragmatic first step and acknowledged this limitation:

Methods, lines 631-634: *“These outcomes were chosen because effective patient-centered communication is a well-established component of clinical effectiveness^{14,50}, and co-design participants emphasized that time efficiency and care coordination were essential prerequisites for LLM adoption within their high-workload environments.⁴⁸”*

Discussion, lines 400-405: *“Fifth, our trial successfully establishes effectiveness within current self-referral practice constraints, providing the essential foundation for subsequent community-level studies; a home-based pre-consultation workflow represents the future integration for exploration. Finally, this multi-center RCT establishes the operational foundation for longitudinal studies of clinical impact. Future longer-term tracking of clinical outcomes is warranted to assess the sustainability.”*

4. Although the authors acknowledge potential biases arising from using local medical dialogue datasets, they do not thoroughly investigate this critical issue. Their analysis lacks depth regarding how these biases might affect clinical decision-making or patient outcomes.

Response: We thank the reviewer for emphasizing this critical dimension. Our ablation study was explicitly designed to quantify how biases in local medical dialogue datasets propagate into clinical decision-making. The data-tuned model (co-designed base + local dialogue fine-tuning) exhibited severe degradation versus co-designed PreA across all domains ($p < 0.001$): history-taking (without data-tuned 4.56 ± 0.65 versus data-tuned 3.86 ± 0.81 ; $P < 0.001$; Extended Data Fig. 1b), diagnosis (without data-tuned 4.67 ± 0.55 versus data-tuned 2.47 ± 1.44 ; $P < 0.001$), and testing order (without data-tuned 4.23 ± 1.09 versus data-tuned 2.21 ± 1.12 ; $P < 0.001$). The local data-tuned model exhibited suboptimal adherence to diagnostic guidelines, failing to provide diagnoses (30.0%) or suggest testing (39.3%) when needed. Supplementary Table 5 demonstrates how local data codifies systemic deficiencies: the omission of guideline-recommended history elements and demographic elements (e.g., patient age and sex); failure to provide appropriate tests and diagnoses; and an adopted unfriendly tone.

Additionally, we clarified that ethical constraints prohibited deploying the data-tuned model in our RCT. Our ethical committee disapproved it because its inferior performance (established in the ablation study) posed unacceptable patient risks and a violation of clinical documentation standards (non-adherence to guidelines). However, our analysis confirmed consistency between simulated and real-world outcomes: PreA's performance with real patients (from RCT) aligned with its virtual patient performance (ablation study) across matched domains (Fig. 4b and Extended Data Fig. 1b). This validates the ablation study as a rigorous proxy for real-world bias impacts.

In this revision, we have amended Methods to clarify these design choices:

Methods, lines 578-583: *“The virtual patient experiment utilized 300 unused patient profiles to evaluate clinical decision impacts (history-taking, diagnosis, test ordering). Referral reports from both variants were blindly evaluated by the same expert panels using validated 5-point Likert scales for completeness, appropriateness, and clinical relevance as in the model development (Supplementary Table 1).”*

Response to referees

Reviewer #1 (Remarks to the Author):

Remarks to the author:

The authors have adequately addressed most of my initial concerns. The response is adequate and comprehensive

This paper evaluated a LLM (PreA), which was co-designed with many stakeholders, including patients, to enhance its effectiveness and the operational efficiency in high-volume specialist hospital settings. The pragmatic RCT showed that patient care and operational efficiency augmented by this LLM (PreA) was superior to usual clinical practice, and appeared to more adequately "prepare" patients for specialist consultation.

Response: Thank you for your positive assessment of our revisions and for recognizing the value of our work.

Reviewer #3 (Remarks to the Author):

This is a single-blinded, pragmatic trial design, which cannot objectively assess the differences in the primary outcome. This trial was single-blinded: While the patients knew their group assignments (PreA-only, PreA-human, or No-PreA), the physicians were uninformed about the PreA-intervention groups (PreA-only or PreA-human), and the researchers were also unaware of the assignments. All evaluators were masked throughout the study.

The primary outcome was the duration of the medical consultation between patients and physicians, defined as the time elapsed from when patients started conversing with physicians to the end of the consultation. With patients knowing their group assignments, this may cause bias.

Response: We thank the reviewer for bringing this important methodological consideration to our attention. The reviewer is correct that patients were aware of their group assignment, which, in principle, could introduce performance bias by altering communication behavior during consultations and detection bias (systematic error in outcome measurement due to knowledge of treatment allocation). This design was intentional and reflects the pragmatic nature of our trial (Ford & Norrie, 2016), as patients would naturally know in real-world practice whether they had used a pre-consultation tool like PreA, which could potentially affect their interaction style. Our goal was to evaluate the intervention under conditions mirroring actual clinical implementation.

That said, we took three steps to assess and mitigate potential bias. To address detection bias, consultation duration was automatically recorded via the electronic platforms, with data analysts blinded to group assignments. To assess performance bias, blinded evaluators analyzed physician clinical notes, which revealed no significant differences between groups, suggesting minimal systematic alteration in patient communication affecting clinical documentation. Furthermore, we observed concordance across multiple outcome measures, including patient-reported experiences and physician-reported care coordination, strengthening the validity of the consultation time reductions as genuine efficiency gains.

Additionally, contextual evidence suggests minimal systematic manipulation: all patients in the PreA and control groups paid standard hospital visit fees and experienced similarly long waiting times (lines 218-221), making it unlikely that PreA-group patients would intentionally shorten their consultations. Furthermore, the consultation duration patterns for the No-PreA group aligned with established clinical practice as confirmed by hospital staff review of pre-trial records.

In this revision, we recognize the inherent limitation of patient unblinding but observe that the consistency across objective, blinded, and subjective measures supports the strength of our results. However, we understand your concern and acknowledge that additional studies are necessary to confirm these findings. We have revised the Limitations section accordingly:

Discussion, lines 389-407: “Third, the single-blinded, pragmatic trial design, while reflecting real-world conditions where patients would naturally know their pre-consultation experience, introduces potential performance bias as patients were aware of their group assignment. However, several factors mitigate this concern: the concordance of findings across multiple outcome assessments, the absence of significant differences in clinical documentation between groups, and the alignment of control group consultation times with established practice patterns.... Finally, our findings should be considered preliminary evidence in support of integrating patient-facing LLMs into hospital outpatient workflows. More extensive multi-center trials with longer follow-up periods are needed to establish sustained clinical benefits, evaluate cost-effectiveness, and validate implementation across diverse healthcare systems.”

References

Ford I, Norrie J. Pragmatic trials. *New England Journal of Medicine*. 2016;375(5):454-63.